# OS-W2S: An Automatic Labeling Engine for Language-Guided Open-Set Aerial Object Detection

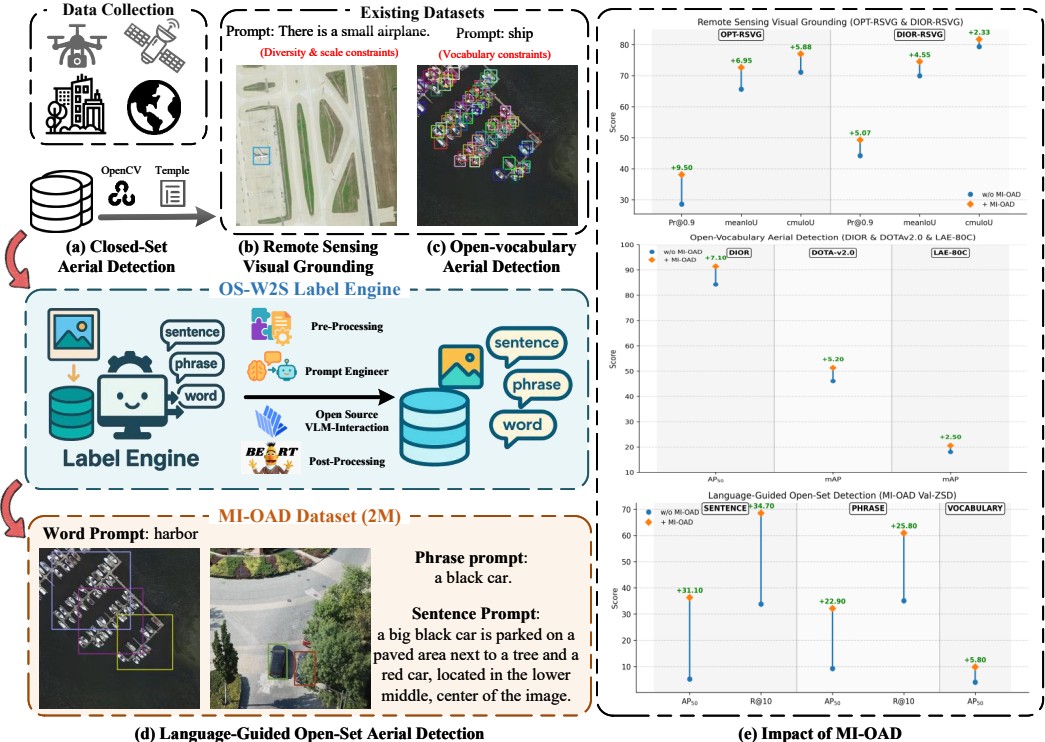

Figure 1: OS-W2S Label Engine and MI-OAD Dataset Construction for Language-Guided Open-Set Aerial Detection. **Left**: The OS-W2S Label Engine pipeline automatically expands existing aerial detection datasets with multi-granularity textual captions ranging from words to sentences, enabling the construction of MI-OAD. Unlike existing tasks, language-guided open-set aerial detection supports multi-granularity language guidance (word, phrase, and sentence levels), making it more aligned with real-world application requirements. **Right**: Performance improvements achieved by MI-OAD across three representative aerial detection tasks: Remote Sensing Visual Grounding, Open-Vocabulary Aerial Detection, and Language-Guided Open-Set Detection, showing substantial gains over baselines without MI-OAD.

## Abstract

In recent years, language-guided open-set aerial object detection has gained significant attention due to its better alignment with real-world application needs. However, due to limited datasets, most existing language-guided methods primarily focus on vocabulary-level descriptions, which fail to meet the demands of fine-grained open-world detection. To address this limitation, we propose constructing a large-scale language-guided open-set aerial detection dataset, encompassing three levels of language guidance: from words to phrases, and ultimately to sentences. Centered around an open-source large vision-language model and integrating image-operation-based preprocessing with BERT-based postprocessing,

we present the **OS-W2S Label Engine**, an automatic annotation pipeline capable of handling diverse scene annotations for aerial images. Using this label engine, we expand existing aerial detection datasets with rich textual annotations and construct a novel benchmark dataset, called Multi-instance Open-set Aerial Dataset (**MI-OAD**), addressing the limitations of current remote sensing grounding data and enabling effective language-guided open-set aerial detection. Specifically, MI-OAD contains 163,023 images and 2 million image-caption pairs, with multiple instances per caption, approximately 40 times larger than comparable datasets. To demonstrate the effectiveness and quality of MI-OAD, we evaluate three representative tasks: language-guided open-set aerial detection, open-vocabulary aerial detection (OVAD), and remote sensing visual grounding (RSVG). On language-guided open-set aerial detection, training on MI-OAD lifts Grounding DINO by +31.1 $AP_{50}$ and +34.7 Recall@10 with sentence-level inputs under zero-shot transfer. Moreover, using MI-OAD for pre-training yields state-of-the-art performance on multiple existing OVAD and RSVG benchmarks, validating both the effectiveness of the dataset and the high quality of its OS-W2S annotations. More details are available at https://anonymous.4open.science/r/MI-OAD.

# 1 INTRODUCTION

Aerial object detection is fundamental to accurately identifying and localizing objects of interest in aerial imagery (Gui et al., 2024). It plays a crucial role in various applications, such as environmental monitoring, urban planning, and rescue operations (Allen et al., 2024; Weng, 2012; Zhao et al., 2024). Most existing aerial detectors are designed to address the inherent challenges of aerial images but are limited to predefined categories and specific scenarios, which defines them as closed-set detectors. However, as drone and satellite technologies advance, the growing need for versatile applications makes closed-set detectors inadequate for real-world scenarios.

Recently, several studies (Li et al., 2023; Zang et al., 2024; Pan et al., 2024; Wei et al., 2025) have explored open-vocabulary aerial detection, which aims to establish relationships between instance region features and corresponding textual category embeddings to overcome the limitations of closed-set detectors. For example, CastDet (Li et al., 2023) adopts a multi-teacher design to leverage superior image-text alignment capabilities from pre-trained VLMs, while OVA-Det (Wei et al., 2025) uses text guidance to further enhance image-text alignment. From a dataset perspective, LAE-DINO (Pan et al., 2024) employs VLMs to expand detectable categories, thereby increasing category diversity and enriching semantic content. Although these methods successfully equip models with vocabulary-guided detection capabilities, they still operate within a framework of discrete categories and remain insensitive to fine-grained textual descriptions. For instance, when given the query "a white car near the green taxi," existing models cannot effectively parse spatial relationships and only focus on individual nouns (car, taxi) in the description.

Language-guided open-set object detection has emerged as a promising solution that can accept arbitrary textual inputs and detect corresponding instances in images. This approach has achieved remarkable progress in natural scenes (Liu et al., 2024; Ren et al., 2024b;a). We observe that these successes are predominantly driven by abundant grounding data. For instance, Grounding DINOv1.5 (Ren et al., 2024b) provides robust language-guided open-set detection capability by training on over 20 million grounding samples, while DINO-X (Ren et al., 2024a) leverages over 100 million data samples. In stark contrast, aerial grounding data remains critically scarce. Only a few attempts (Sun et al., 2022; Zhan et al., 2023; Li et al., 2024) have been made to construct remote sensing visual grounding (RSVG) datasets by annotating detection data with captions, yet these datasets suffer from several limitations: *1) Lack of scene diversity*: Existing RSVG datasets predominantly rely on the DIOR dataset and apply restrictive conditions, such as limiting images to fewer than five objects per category. Although such constraints help ensure annotation quality, they also exclude complex scenes that are essential for training robust language-guided open-set detectors. *2) Limited caption diversity*: Current RSVG datasets predominantly employ fixed templates and rule-based attribute extraction, resulting in a lack of flexibility and diversity in textual descriptions. *3) Single-instance annotation*: Existing RSVG datasets focus exclusively on referring expression comprehension tasks, associating one caption with a single instance. However, practical applica-

tions often require models to retrieve all instances matching a given caption, where the number of retrieved instances should be determined by the caption's specificity rather than being artificially limited to a single result. *4) Limited dataset scale*: The largest available RSVG dataset comprises approximately 25,452 images and 48,952 image-caption pairs. Compared to successful natural-image-based language-guided open-set detectors (e.g., Grounding DINO v1.5 with 20M samples, DINO-X with 100M samples), existing aerial RSVG datasets are severely constrained in scale. This substantial gap in data scale critically restricts the development of robust language-guided open-set aerial detection models.

To bridge this gap, in this paper, we aim to lay the data foundation for language-guided open-set aerial object detection. Specifically, we propose the *OS-W2S* Label Engine, an automatic annotation pipeline capable of handling diverse scene annotations for aerial images. It is based on an open-source vision-language model, image-operate-based preprocessing, and BERT-based postprocessing. Using this label engine, we construct a novel large-scale benchmark dataset, called MI-OAD, to overcome the limitations of current RSVG data.

Key aspects include: *1) Scene Diversity:* We collected data from eight representative aerial detection datasets covering diverse scenarios from various altitudes and viewpoints. We also introduced pre-processing and post-processing steps to enable the pipeline to effectively handle arbitrary aerial scenes without filtering out complex scenarios, thereby preserving comprehensive scene diversity. *2) Caption Diversity:* Leveraging the robust vision-language capabilities of VLMs, we generate six distinct caption types per instance with varying levels of detail based on attribute combinations. The resulting captions average 10.61 words in length and encompass rich attributes including category, color, size, geometry, and both relative and absolute positional information, ensuring comprehensive semantic diversity for different localization requirements. *3) Multi-instance annotation:* Unlike existing RSVG datasets limited to single instance per caption, we match varying numbers of instances to each caption based on its descriptive specificity during post-processing. This design yields flexible caption-instance associations, better aligning with diverse real-world application requirements. *4) Dataset Scale:* Using this label engine, we expanded eight widely used aerial detection datasets, yielding 163,023 images and 2 million image-caption pairs, which is 40 times larger than those available in existing RS grounding datasets.

In summary, our contributions are three-fold: (1) We introduce the OS-W2S Label Engine, an automatic annotation pipeline that lays the data foundation for language-guided open-set aerial object detection and can be executed on a single workstation equipped with eight RTX4090 GPUs. (2) Using this engine, we present MI-OAD, the first benchmark for this task, encompassing 163,023 images and 2 million image-caption pairs with multiple instances per caption, annotated at the word-, phrase-, and sentence-level descriptions. (3) Comprehensive experiments demonstrate that MI-OAD enables significant performance improvements across language-guided open-set aerial detection, open-vocabulary aerial detection, and remote sensing visual grounding tasks.

## 2 OS-W2S LABEL ENGINE

As shown in Fig. 2, the OS-W2S Label Engine consists of the following four components:

**Data Collection.** To construct a high-quality language-guided open-set aerial detection benchmark, we build upon eight representative aerial detection datasets with verified human annotations (Li et al., 2020b; Xia et al., 2018; Zhang et al., 2019; Su et al., 2019; Xiao et al., 2015; Pisani et al., 2024; Zhu et al., 2021; Lam et al., 2018). These carefully curated datasets exhibit rich scene diversity due to variations in capturing altitudes and sensing platforms (ranging from satellites to drones), while providing precise detection annotations with accurate instance categories and bounding-box coordinates. To ensure consistency across heterogeneous data sources, we standardized all datasets by cropping images to uniform resolutions and converting annotations to a unified format. This comprehensive data collection and standardization process establishes a robust, quality-assured foundation for our subsequent annotation pipeline.

**Data Preprocessing.** Data preprocessing aims to bridge the domain gap between VLMs trained on natural images and aerial image annotation tasks by providing structured visual input and reliable attribute priors to improve annotation quality. As shown in Figure 2, our preprocessing pipeline is based on two guiding principles for subsequent robust VLM-based annotation: (1) *Reliable attribute*

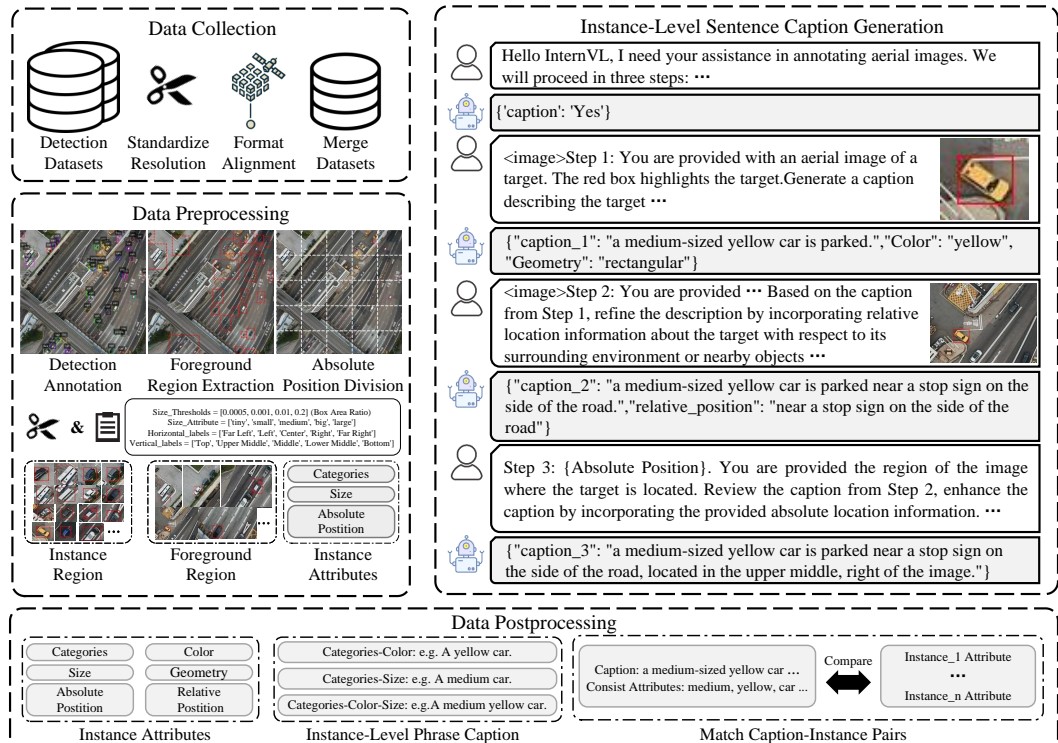

Figure 2: The pipeline of the proposed OS-W2S Label Engine. The labeling process includes four major components: Data Collection, Data Preprocessing, Instance-Level Sentence Caption Generation, and Data Postprocessing. Each aerial image undergoes a comprehensive annotation process involving attribute extraction, caption generation with varying detail levels using a VLM, and precise matching of caption-instance associations based on attribute similarity.

*prior guidance*: Following (Ma et al., 2024; Zhan et al., 2023), we leverage instance attributes as foundational components for diverse caption generation. We focus on six primary attributes: category, size, color, geometric shape, relative position, and absolute position. To ensure annotation reliability and mitigate domain gap effects, we explicitly provide three priors per instance to the VLM: (i) category information from detection annotations, (ii) size calculated as area ratios using rule-based methods, and (iii) absolute position categorized into 25 spatial regions (e.g., Left-Top, Far Right-Bottom). The remaining attributes are dynamically generated by the VLM based on visual content. This strategic incorporation of reliable prior knowledge reduces ambiguity and yields more dependable annotations. (2) *Local visual crop guidance*: Appropriate cropping helps the VLM focus on correct targets and provides more fine-grained information. However, excessive zooming may omit crucial contextual information, while insufficient zooming may miss important object details. After iterative experimentation, we adopted a two-stage cropping strategy to optimally balance fine details and relevant context: (i) Instance regions: We crop sub-images based on detection bounding boxes to ensure the VLM focuses on specific targets while capturing sufficient fine-grained details for accurate instance attribute generation. (ii) Foreground regions: Given the dense instance distribution and extensive background interference in aerial imagery, we employ a foreground-extraction algorithm (Algorithm 2 in Appendix) to provide essential local context, enabling accurate relative position generation while minimizing background noise interference.

**Instance-Level Sentence Caption Generation.** This step leverages VLM interaction to generate the remaining attributes and captions with varying levels of detail for each instance. The interaction with the VLM for each instance is structured into four rounds: (i) Workflow initialization: Introduction of the overall annotation workflow to the VLM. (ii) Color and geometry attribute generation: By providing the instance region with category and size priors, we ensure the VLM focuses on the specific instance, enabling accurate generation of color and geometry attributes along with a self-descriptive caption while minimizing errors or hallucinations. (iii) Relative position attribute

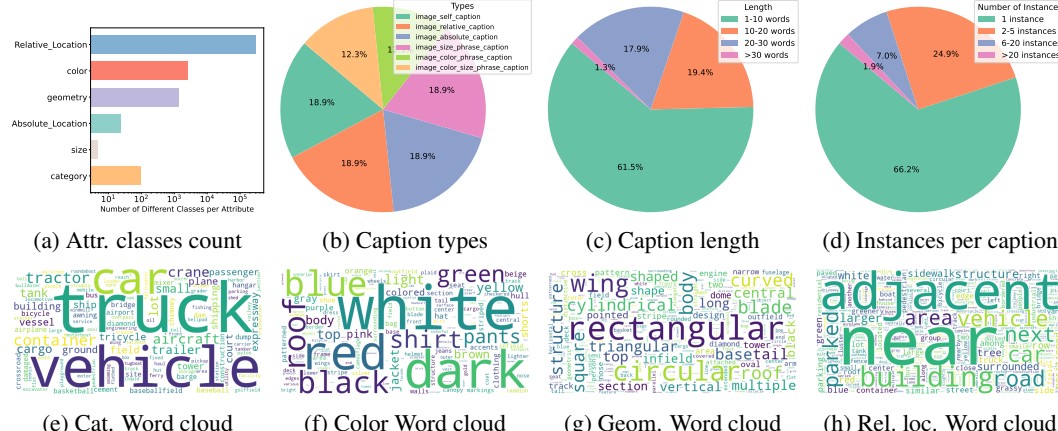

(a) Attr. classes count     (b) Caption types     (c) Caption length     (d) Instances per caption

(e) Cat. Word cloud     (f) Color Word cloud     (g) Geom. Word cloud     (h) Rel. loc. Word cloud

Figure 3: Statistical analysis and visualization of the MI-OAD dataset. (a) The number of distinct values per attribute type, highlighting attribute diversity. (b) Distribution of caption types, showing that caption types are evenly distributed. (c) Distribution of caption lengths, reflecting semantic richness. (d) Distribution of the number of instances per caption, demonstrating that our dataset includes both single-instance and multi-instance correspondences, making it more suitable for practical applications. (e) Word cloud visualization of categories sourced from the collected detection datasets. (f)-(h) Word cloud visualizations illustrating diverse semantic expressions for color, geometry, and relative position attributes generated by the VLM.

generation: We provide the foreground region containing the target instance's immediate surroundings, with the target highlighted by a red bounding box. This ensures the VLM correctly identifies which object to describe and can accurately infer relative position attributes to generate corresponding captions, maintaining clear focus while reducing confusion from broader background noise. (iv) Absolute position integration: We provide the absolute position attribute to the VLM, prompting it to integrate this spatial information into the existing caption, thereby generating a comprehensive caption that fully reflects the instance's spatial context. Consequently, each instance is annotated with three distinct sentence captions and all six attributes. We selected InternVL-2.5-38B-AWQ as our VLM based on its superior capability and practical efficiency (see Appendix C for details).

**Data Postprocessing.** We first generate three phrase-level captions per instance using combinations of category, color, and size attributes. Since a single caption may describe multiple instances within an image, we establish caption-instance associations by computing the compositional attribute similarity between captions and instances. This approach enables our dataset to transcend the conventional REC limitation of one-to-one caption-instance mapping, satisfying both precise and approximate localization requirements in practical applications. Attribute similarity is computed using Sentence-BERT (Reimers & Gurevych, 2019).

## 3 MI-OAD DATASET

Using the OS-W2S Label Engine, we created MI-OAD, a large-scale dataset for language-guided open-set aerial object detection. This dataset comprises 163,023 images and 2 million image-caption pairs, encompassing three levels of language guidance: vocabulary-level, phrase-level, and sentence-level descriptions. With an average caption length of 10.61 words, the dataset provides rich semantic information for comprehensive aerial scene understanding. Detailed statistical analysis is presented in Fig. 3. Benefiting from the systematic design of the OS-W2S Label Engine, the MI-OAD dataset effectively addresses the limitations of existing RSVG datasets and establishes the first benchmark for language-guided open-set aerial object detection.

**Scene Diversity:** We made two efforts to ensure scene diversity. First, we collected data from eight representative detection datasets, which include images captured from various altitudes and viewpoints using drones and satellites. Second, we carefully designed preprocessing and postprocessing

pipelines to ensure the label engine can handle arbitrary aerial scenes, thereby eliminating the need to filter out complex scenes and preserving comprehensive scene diversity.

**Caption Diversity:** To comprehensively cover real-world application requirements, we generate six captions per instance: three sentence caption types and three phrase caption types, each varying in detail based on attribute combinations. The sentence captions provide detailed instance descriptions suitable for precise localization: self sentence captions describe category, size, color, and geometric attributes; relative sentence captions additionally incorporate relative positional information; and absolute sentence captions further include absolute positional information. Additionally, three types of phrase captions constructed from combinations of category, color, and size attributes support approximate localization. Fig.3 provides comprehensive visual analysis that further demonstrates the caption diversity within our dataset. Fig.3a presents the number of distinct expressions for each attribute, highlighting the rich diversity in attributes (relative location, color, and geometry) generated by the VLM. Fig.3b illustrates the distribution of caption types, showing that after applying the sampling strategy described in Section5.1, the caption types are evenly distributed across the dataset. Fig.3(f)-(h) present word cloud analyses that visually demonstrate the diversity of these VLM-generated attributes. Furthermore, Fig.3c depicts the distribution of caption lengths, illustrating the richness of our descriptions. Collectively, these analyses underscore the comprehensive caption diversity within our dataset.

**Multi-instance Annotation:** To better align with real-world applications requiring both precise and approximate localization, we construct caption-instance associations by comparing the attributes of captions and instances, allowing each caption to correspond to different numbers of instances based on the specificity of the descriptive information it contains. As shown in Fig. 3d, 66.2% of captions correspond to a single instance, demonstrating that the generated captions effectively support precise localization even in complex scenes. The remaining captions, which involve multiple instances, fulfill the requirements for approximate localization.

**Dataset Scale:** To ensure data quality, we deliberately selected eight representative aerial detection datasets with rigorous annotation processes and high-quality human annotations, and expanded them with rich textual descriptions. The resulting MI-OAD dataset contains 163,023 images and 2 million image-caption pairs, making it 40 times larger than the existing RSVG dataset. Notably, the OS-W2S Label Engine is easily reproducible, enabling researchers to seamlessly incorporate additional datasets for enhanced scale.

## 4    QUALITY CONTROL ANALYSIS

To ensure the reliability of annotations produced by the *OS-W2S Label Engine*, we adopt three complementary safeguards: *(i) Curated sources and preprocessing priors.* We start from widely used aerial detection datasets whose bounding boxes and category labels were manually verified. These well-curated sources allow us to inherit precise detection annotations, enabling the preprocessing stage to generate reliable guidance priors (attribute priors and local visual cropping) for subsequent annotations. This approach provides adequate context for accurate VLM-generated annotations, thereby reducing the impact of VLMs' domain gap and mitigating hallucination risks. *(ii) Structured prompts and output validation.* Through carefully crafted prompts, we enable the VLM to fully comprehend task requirements and generate outputs in a predefined, machine-parsable format. The returned text is validated using regular expression matching, ensuring a 100% parsing success rate. *(iii) Benchmark construction with manual review.* To establish MI-OAD as a reliable benchmark, we construct a high-quality test set through rigorous manual review. Specifically, we grouped the MI-OAD validation image-caption pairs by category and had five experts manually select 10,000 high-quality pairs (approximately 100 per category) to form the MI-OAD test set.

Beyond these safeguards, we conducted a quantitative evaluation of the generated attributes. Specifically, we sampled 300 images (1,765 instances) from the MI-OAD dataset and employed two powerful independent models: a stronger open-source VLM (InternVL3-78B) and a leading closed-source model (GPT-4o-mini) to verify the accuracy of three VLM-generated attributes (color, geometry, and relative position) in matching the corresponding instance images. InternVL3-78B attained 98.98% (color), 99.21% (geometry), and 97.90% (relative position), while GPT-4o-mini achieved 96.88%, 94.39%, and 96.92%, respectively. These consistently high accuracy rates provide quantitative evidence of our annotation quality.

## 5 EXPERIMENTS

In this section, we validate the effectiveness of our proposed MI-OAD dataset from three key aspects: (1) We validate that MI-OAD can equip models with language-guided open-set aerial detection capabilities and establish a benchmark for this task. (2) We demonstrate MI-OAD's effectiveness in enhancing two existing tasks: open-vocabulary aerial detection and remote sensing visual grounding. (3) We verify the necessity of a large-scale dataset like MI-OAD for advancing language-guided open-set aerial detection. Due to space constraints, the analysis of dataset scale impact (point 3) and detailed descriptions of the label engine, category split, experimental settings, and qualitative examples are provided in the appendix.

### 5.1 MI-OAD DATASET SPLIT AND SAMPLE

**Base and Novel Classes Split.** To rigorously evaluate models' zero-shot transfer capabilities, we split the 100 categories into 75 base classes and 25 novel classes, and the base/novel split follows prior work (Zang et al., 2024).

**Data Split.** To fully leverage the available data while preserving the original datasets partition structure, we first merge the train and test splits of all eight collection datasets. Then, We select samples containing only base class instances to form the pre-training set (P-Set), while using the entire merged dataset as the fine-tuning set (FT-Set). The validation splits are processed using the same principle: samples containing novel classes are sampled to form Val-ZSD, while the complete validation set serves as Val-FT. Thus, we train on P-Set and evaluate on Val-ZSD to assess zero-shot transfer capabilities, while models fine-tuned on FT-Set are evaluated on Val-FT to measure overall detection performance.

**Sampling Strategy and Experimental Data Statistics.** After processing with the OS-W2S Label Engine, each image corresponds to multiple captions. Considering the potential overfitting risks from image reuse and substantial computational resource requirements, we implemented a caption sampling strategy. Specifically, for each image, we categorized captions by caption type and corresponding object categories, then performed uniform sampling across both caption types and object categories to form image-caption pairs, ensuring dataset diversity. Ultimately, the MI-OAD dataset comprises approximately 2 million image-caption pairs and 163,023 detection annotations. The P-Set comprises 0.56M image-caption pairs and 68,243 detection annotations. The FT-Set includes 1.40M pairs and 128,019 annotations. For validation: Val-ZSD provides about 0.12M pairs and 16,992 detection annotations for zero-shot evaluation, whereas Val-FT contains roughly 0.38M pairs and 35,004 annotations for conventional detection performance assessment.

### 5.2 EVALUATION SETUPS

To comprehensively evaluate language-guided open-set detection capability, we propose three evaluation protocols simulating real-world scenarios: vocabulary-level detection, phrase-level grounding, and sentence-level grounding, corresponding to varying levels of linguistic detail. Additionally, we define three evaluation setups to assess detection performance under different constraints: (1) zero-shot transfer to novel classes without domain adaptation, (2) zero-shot transfer to novel classes with domain adaptation, and (3) fine-tuned evaluation. Domain adaptation refers to fine-tuning detectors originally trained only on natural images using the MI-OAD P-Set. Fine-tuned evaluation refers to fine-tuning the model on the MI-OAD FT-Set.

### 5.3 LANGUAGE-GUIDED OPEN-SET AERIAL OBJECT DETECTION RESULTS

Table 1 presents a comprehensive evaluation of two representative language-guided open-set detectors—YOLO-World (YOLOv8-L) and Grounding DINO (Swin-T)—on MI-OAD across three granularities: vocabulary-level detection, phrase-level grounding, and sentence-level grounding. We examine three progressive training conditions to showcase how MI-OAD addresses key challenges in language-guided open-set aerial detection: enabling effective domain adaptation, supporting zero-shot transfer to novel classes, and achieving strong generalized detection performance. Since validation and test sets exhibit same trends, , the following analysis focuses on the *Validation Set*.

Table 1: Performance evaluation on MI-OAD across vocabulary-level detection, phrase-level grounding, and sentence-level grounding tasks. Validation results assess zero-shot transfer (Val-ZSD) and overall detection (Val-FT) capabilities. Test set provides manually verified grounding results only, as detection annotations are pre-verified. Three training conditions are examined: zero-shot without adaptation, zero-shot with P-Set domain adaptation, and full fine-tuning on FT-Set.

| Method | Detection | | Phrase Grounding | | | | Sentence Grounding | | | |
|---|---|---|---|---|---|---|---|---|---|---|
| | $AP_{50}$ | R@100 | $AP_{50}$ | R@1 | R@10 | R@100 | $AP_{50}$ | R@1 | R@10 | R@100 |
| *MI-OAD Validation Set* | | | | | | | | | | |
| *Zero-shot transfer with novel classes (w/o domain adaptation)* | | | | | | | | | | |
| YOLO-World (Cheng et al., 2024) | 3.2 | 37.1 | 3.8 | 6.8 | 25.0 | 34.4 | 1.4 | 4.3 | 16.9 | 24.6 |
| Grounding DINO (Liu et al., 2024) | 4.0 | 49.6 | 9.2 | 10.7 | 35.1 | 50.4 | 5.2 | 10.3 | 33.8 | 42.9 |
| *Zero-shot transfer with novel classes (w/ domain adaptation on P-Set)* | | | | | | | | | | |
| YOLO-World | 5.3 | 30.6 | 18.0 | 18.3 | 43.5 | 55.9 | 15.9 | 19.1 | 44.9 | 57.1 |
| Grounding DINO | 9.8 | 69.8 | 32.1 | 24.1 | 60.9 | 80.9 | 36.3 | 35.1 | 68.5 | 82.7 |
| *Full fine-tuning on FT-Set* | | | | | | | | | | |
| YOLO-World | 39.6 | 58.0 | 51.6 | 32.9 | 69.9 | 86.4 | 47.6 | 36.1 | 71.1 | 86.9 |
| Grounding DINO | 37.1 | 70.1 | 57.8 | 35.2 | 74.4 | 91.5 | 56.4 | 44.1 | 78.0 | 90.3 |
| *MI-OAD Test Set* | | | | | | | | | | |
| *Zero-shot transfer with novel classes (w/ domain adaptation on P-Set)* | | | | | | | | | | |
| YOLO-World | - | - | 19.5 | 18.6 | 42.3 | 55.0 | 16.4 | 19.7 | 43.7 | 55.4 |
| Grounding DINO | - | - | 33.2 | 24.4 | 60.3 | 81.1 | 37.6 | 35.4 | 68.8 | 82.7 |
| *Full fine-tuning on FT-Set* | | | | | | | | | | |
| YOLO-World | - | - | 52.7 | 34.2 | 70.8 | 87.8 | 47.9 | 36.0 | 71.4 | 86.6 |
| Grounding DINO | - | - | 58.3 | 35.7 | 75.3 | 92.2 | 57.3 | 44.0 | 78.0 | 89.7 |

**Zero-shot Transfer without Domain Adaptation.** Directly applying models trained solely on natural images to aerial imagery exposes pronounced domain gaps. YOLO-World attains only 1.4% $AP_{50}$ for sentence-level grounding, while Grounding DINO performs slightly better at 5.2% $AP_{50}$ yet still exhibits severe limitations. These results confirm that natural-image language-guided open-set detectors cannot transfer reliably to aerial domains, underscoring the need for dedicated, large-scale aerial grounding datasets such as MI-OAD.

**Zero-Shot Transfer with Domain Adaptation.** Introducing domain adaptation by training on the MI-OAD P-Set yields substantial gains for both methods. For Grounding DINO, detection $AP_{50}$ rises from 4.0% to 9.8% and sentence-level $AP_{50}$ jumps from 5.2% to 36.3% (+31.1 percentage points). Similarly, YOLO-World's sentence-level $AP_{50}$ improves from 1.4% to 15.9%. These improvements highlight MI-OAD's effectiveness in bridging the domain gap and demonstrate that MI-OAD equips detectors with strong zero-shot performance on novel classes.

**Full Fine-Tuning.** After fine-tuning on the MI-OAD FT-Set, both models achieve markedly stronger results. Grounding DINO reaches $AP_{50}$ values of 37.1% for detection, 57.8% for phrase-level grounding, and 56.4% for sentence-level grounding. These findings show that MI-OAD provides a solid data foundation for advancing language-guided open-set aerial object detection and confirm the importance of large-scale grounding data with rich textual annotations.

## 5.4 Performance on Remote-Sensing Visual Grounding

To further assess the effectiveness of MI-OAD for remote-sensing visual grounding, Table 2 reports Grounding DINO's performance on OPT-RSVG and DIOR-RSVG under two training paradigms: (i) training solely on the respective RSVG training sets and (ii) incorporating MI-OAD as additional training data.

The results demonstrate substantial improvements by incorporating MI-OAD. On OPT-RSVG, Grounding DINO's Pr@0.9 increases from 28.6% to 38.1%, while mean IoU improves from 65.7% to 72.6%. This surpasses the previous state-of-the-art LPVA by 15.7 points (Pr@0.8) and 6.4 points (meanIoU), establishing new benchmarks on OPT-RSVG. Similarly, on DIOR-RSVG, Pr@0.9 rises

Table 2: Comparison with state-of-the-art methods on the OPT-RSVG and DIOR-RSVG test sets (English version). "Gain over GD" compares against Grounding DINO trained on each set; "Gain over LPVA" compares against the prior SOTA method.

| Method | OPT-RSVG | | | | | | | DIOR-RSVG | | | | | | |
|---|---|---|---|---|---|---|---|---|---|---|---|---|---|---|
| | Pr@0.5 | Pr@0.6 | Pr@0.7 | Pr@0.8 | Pr@0.9 | meanIoU | cmuIoU | Pr@0.5 | Pr@0.6 | Pr@0.7 | Pr@0.8 | Pr@0.9 | meanIoU | cmuIoU |
| *One-stage* | | | | | | | | | | | | | | |
| ZSGNet (ICCV'19) (Sadhu et al., 2019) | 48.64 | 47.32 | 43.85 | 27.69 | 6.33 | 43.01 | 47.71 | 51.67 | 48.13 | 42.30 | 32.41 | 10.15 | 44.12 | 51.65 |
| FAOA (ICCV'19) (Yang et al., 2019b) | 68.13 | 64.30 | 57.15 | 41.83 | 15.33 | 58.79 | 65.20 | 67.21 | 64.18 | 59.23 | 50.87 | 34.44 | 59.76 | 63.14 |
| ReSC (ECCV'20) (Yang et al., 2020) | 69.12 | 64.63 | 58.20 | 43.01 | 14.85 | 60.18 | 65.84 | 72.71 | 68.92 | 63.01 | 53.70 | 33.37 | 64.24 | 68.10 |
| LBYL-Net (CVPR'21) (Huang et al., 2021) | 70.22 | 65.39 | 58.65 | 37.54 | 9.46 | 60.57 | 70.28 | 73.78 | 69.22 | 65.56 | 47.89 | 15.69 | 65.92 | 76.37 |
| *Transformer-based* | | | | | | | | | | | | | | |
| TransVG (CVPR'21) (Deng et al., 2021) | 69.96 | 64.17 | 54.68 | 38.01 | 12.75 | 59.80 | 69.31 | 72.41 | 67.38 | 60.05 | 49.10 | 27.84 | 63.56 | 76.27 |
| QRNet (CVPR'22) (Ye et al., 2022) | 72.03 | 65.94 | 56.90 | 40.70 | 13.35 | 60.82 | 75.39 | 75.84 | 70.82 | 62.27 | 49.63 | 25.69 | 66.80 | 83.02 |
| VLTGV (R-50) (CVPR'22) (Yang et al., 2022) | 71.84 | 66.54 | 57.79 | 41.63 | 14.62 | 60.78 | 70.69 | 69.41 | 65.16 | 58.44 | 46.56 | 24.37 | 59.96 | 71.97 |
| VLTGV (R-101) (CVPR'22) (Yang et al., 2022) | 73.50 | 68.13 | 59.93 | 43.45 | 15.31 | 62.48 | 73.86 | 75.79 | 72.22 | 66.33 | 55.17 | 33.11 | 66.32 | 77.85 |
| MGVLF (TGRS'23) (Zhan et al., 2023) | 72.19 | 66.86 | 58.02 | 42.51 | 15.30 | 61.51 | 71.80 | 75.98 | 72.06 | 65.23 | 54.89 | 35.65 | 67.48 | 78.63 |
| LPVA (TGRS'24) (Li et al., 2024) | 78.03 | 73.32 | 62.22 | 49.60 | 25.61 | 66.20 | 76.30 | 82.27 | 77.44 | 72.25 | 60.98 | 39.55 | 72.35 | **85.11** |
| Grounding DINO (Train on each train set) | 75.73 | 72.62 | 66.30 | 53.29 | 28.63 | 65.66 | 71.12 | 77.85 | 75.69 | 71.14 | 62.65 | 44.19 | 69.96 | 79.36 |
| Grounding DINO (+MI-OAD) | 82.62 | 80.83 | 76.59 | 65.26 | 38.13 | 72.61 | 77.00 | 82.46 | 80.92 | 77.43 | 69.20 | 49.26 | 74.51 | 81.69 |
| Gain over GD (Train on each train set) | +6.89 | +8.21 | +10.29 | +11.97 | +9.50 | +6.95 | +5.88 | +4.61 | +5.23 | +6.29 | +6.55 | +5.07 | +4.55 | +2.33 |
| Gain over LPVA (SOTA) | +4.59 | +7.51 | +14.37 | +15.66 | +12.52 | +6.41 | +0.70 | +0.19 | +3.48 | +5.18 | +8.22 | +9.71 | +2.16 | -3.42 |

Table 3: Open-vocabulary aerial detection performance on DIOR, DOTA-v2.0, and LAE-80C benchmarks. The first row shows results from the original paper using 4×A100 GPUs. Results marked with an asterisk (*) are from our reimplementation using 32×RTX 4090 GPUs with default hyperparameters.

| Method | Training Data | DIOR $AP_{50}$ | DOTA-v2.0 mAP | LAE-80C mAP |
|---|---|---|---|---|
| LAE-DINO (Pan et al., 2024) | LAE-1M | 85.5 | 46.8 | 20.2 |
| LAE-DINO* | LAE-1M | 84.3 | 46.1 | 18.0 |
| LAE-DINO* | + MI-OAD | **91.4** (+7.1) | **51.3** (+5.2) | **20.5** (+2.5) |

from 44.2% to 49.3% and mean IoU from 70.0% to 74.5%, outperforming LPVA by 9.71 and 2.16 points, respectively. These results confirm that MI-OAD significantly enhances grounding capabilities, especially at high-IoU thresholds where precise localization is critical. Consistent gains across both datasets reveal a clear scaling effect: larger and more diverse training data yield more robust and better-generalizing models. This underscores the value of both our OS-W2S Label Engine for scalable annotation and the MI-OAD dataset for advancing remote-sensing visual grounding.

## 5.5 PERFORMANCE ON OPEN-VOCABULARY AERIAL DETECTION

To further validate MI-OAD's value, we evaluate open-vocabulary aerial detection with LAE-DINO (Pan et al., 2024). As shown in Table 3, incorporating MI-OAD during pretraining consistently improves performance across all three benchmarks. Relative to training on LAE-1M alone, MI-OAD pretraining yields gains of +7.1 points on DIOR $AP_{50}$ (from 84.3 to 91.4), +5.2 points on DOTA-v2.0 mAP (from 46.1 to 51.3), and +2.5 points on LAE-80C mAP (from 18.0 to 20.5). These consistent improvements across diverse benchmarks highlight MI-OAD's broad applicability and effectiveness for open-vocabulary detection in aerial imagery.

## 6 CONCLUSION

In this paper, we propose the OS-W2S Label Engine, which addresses the scarcity of rich textual grounding data in the aerial domain and establishes a robust data foundation for language-guided open-set aerial detection. Using this pipeline, we introduce MI-OAD, the first benchmark dataset for language-guided open-set aerial detection, containing 163,023 images and 2 million image-caption pairs with multi-granularity descriptions at word, phrase, and sentence levels. Extensive experiments demonstrate that MI-OAD not only enables effective language-guided open-set aerial detection but also achieves state-of-the-art performance on existing open-vocabulary aerial detection and remote sensing visual grounding tasks. Our work provides both a scalable annotation pipeline and a comprehensive benchmark that we hope will accelerate future research in open-set aerial detection.

## 7 REPRODUCIBILITY STATEMENT

To ensure reproducibility of our results, we provide experimental settings in Section 5 of the main text and additional implementation specifics in Appendix E.1. Complete source code, datasets, model configurations, and detailed experimental procedures are made available at `https://anonymous.4open.science/r/MI-OAD`.

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

## A    LARGE LANGUAGE MODEL USAGE STATEMENT

Large Language Models (GPT-5) were used to aid in polishing the manuscript text. Additionally, we employed InternVL-2.5-38B-AWQ for MI-OAD construction, and InternVL-2.5-78B along with GPT-4o-mini for dataset annotation quality assessment.

## B    RELATED WORK

### B.1    LANGUAGE-GUIDED OPEN-SET OBJECT DETECTION

Language-guided open-set object detection, which can accept arbitrary textual inputs and detect corresponding objects based on these descriptions, demonstrates significant potential due to its close alignment with real-world application needs. Several studies (Li et al., 2022; Zhang et al., 2022; Liu et al., 2024; Yao et al., 2023; Cheng et al., 2024; Ren et al., 2024a) have demonstrated the feasibility of language-guided open-set object detection in natural image scenarios. GLIP (Li et al., 2022) established a foundation for language-guided open-set detection by integrating object detection and grounding tasks. Building on this, models such as YOLO-World (Cheng et al., 2024) and the Grounding DINO series (Liu et al., 2024; Ren et al., 2024b;a) have made significant progress. Notably, Grounding DINO v1.5, trained on over 20 million images with grounding annotations, and DINO-X, utilizing over 100 million data samples, both demonstrate exceptional language-guided open-set detection performance, underscoring the crucial role of large-scale grounding data.

Compared to natural image scenarios, the development of language-guided open-set aerial object detection has lagged behind, with grounding data in aerial domains remaining critically scarce. To bridge this gap, this paper aims to establish a comprehensive data foundation for language-guided open-set aerial object detection.

### B.2    OBJECT DETECTION IN AERIAL IMAGERY

Aerial object detection can be broadly divided into two types: closed-set aerial detection and open-vocabulary aerial detection.

Closed-set aerial detection refers to predicting bounding boxes and corresponding categories for objects that have been seen during training. Several studies (Du et al., 2023; Huang et al., 2022; Liu et al., 2021; Li et al., 2020a; Yang et al., 2019a) have primarily focused on addressing the inherent challenges of remote sensing images. For instance, models such as UFPMP-Det (Huang et al., 2022), ClustDet (Yang et al., 2019a), and DMNet (Li et al., 2020a) employ a coarse-to-fine two-stage detection architecture to mitigate significant background interference and effectively detect small, densely distributed objects. However, these models are constrained by predefined training categories, limiting their applicability to specific scenarios in real-world applications.

Open-vocabulary aerial detection represents a step towards meeting the demands of open-world aerial detection. It seeks to eliminate the category limitations inherent in closed-set detection by establishing relationships between image features and category embeddings, rather than simply mapping image features to category indices. Models such as CastDet (Li et al., 2023), DescReg (Zang et al., 2024), and OVA-Det (Wei et al., 2025) leverage the superior image-text alignment capabilities inherited from pre-trained Vision-Language Models (VLMs) to enable open-vocabulary aerial detection capabilities. Additionally, LAE-DINO (Pan et al., 2024) addresses this limitation from a dataset perspective by employing VLMs to expand the detection category set, thereby increasing category diversity and enriching the semantic content of detection text.

Despite these advancements, current research in open-vocabulary aerial detection remains limited to discrete categories and cannot handle the arbitrary textual descriptions that real-world applications demand. Compared to the natural image domain, language-guided open-set object detection in aerial imagery still has significant room for exploration and improvement.

### B.3    VISUAL GROUNDING IN AERIAL IMAGERY

Visual grounding in remote sensing (RSVG) aims to locate objects based on natural language descriptions. Compared to closed-set and open-vocabulary object detection, RSVG can process ar-

bitrary descriptions to identify corresponding targets, offering greater flexibility and suitability for practical applications (Li et al., 2024). However, the inherent challenges in annotating aerial images, which often contain predominantly small objects and substantial background interference, have hindered progress in this field. RSVG remains in its early stages of development, with only three available datasets: RSVG-H (Sun et al., 2022), DIOR-RSVG (Zhan et al., 2023), and OPT-RSVG (Li et al., 2024). Among these, RSVG-H comprises 4,239 RS images paired with 7,933 textual descriptions, each providing precise geographic distances. DIOR-RSVG, based on the DIOR dataset (Li et al., 2020b), makes use of tools such as HSV and OpenCV to extract instance attributes (e.g., geometric and colors) and employs predefined templates to generate 38,320 image-caption pairs. Meanwhile, OPT-RSVG further enriches RSVG scenarios by combining three detection datasets (DIOR, HRRSD (Zhang et al., 2019), and SPCD (Bhartiya, 2019)) and follows the annotation process in Zhan et al. (2023) to produce 25,452 RS images with 48,952 image-caption pairs.

Compared to the abundance of grounding data required by successful language-guided open-set detectors in natural images, the scale of available aerial grounding data remains extremely limited. This scarcity poses a significant barrier for data-driven open-set detection tasks. To address these limitations and lay the data foundation for language-guided open-set aerial object detection, we propose the OS-W2S Label Engine and construct MI-OAD, a large-scale dataset for language-guided open-set aerial detection tasks.

### B.4 Multi-modal Data Labeling Engines for Aerial Detection and Grounding

In open-vocabulary aerial detection, most existing works (Li et al., 2023; Zang et al., 2024; Wei et al., 2025) address data scarcity by leveraging pre-trained vision-language models (VLMs) such as CLIP (Radford et al., 2021) to transfer image-text alignment to the aerial domain. To overcome category limitations in conventional aerial detection datasets, LAE-DINO (Pan et al., 2024) proposed the LAE-Label Engine, which uses SAM to generate object proposals and employs a VLM to assign multiple category names with confidence scores, followed by confidence-based filtering. While this effectively increases category counts, it remains confined to vocabulary-level semantics, providing limited semantic richness.

In the aerial visual grounding domain, DIOR-RSVG (Zhan et al., 2023) and OPT-RSVG (Li et al., 2024) construct their datasets using hand-crafted rules and fixed templates for caption generation. Specifically, they derive a set of attributes (*e.g.*, shape, size, color) from detection labels and low-level tools such as OpenCV, then compose these attributes into captions via pre-defined templates. However, these pipelines are predominantly built on a single base dataset (DIOR), focus exclusively on the referring expression comprehension (REC) task where one caption corresponds to only a single target in an image, and enforce a one-caption-one-instance constraint by filtering out scenes with complex layouts. Consequently, they suffer from: (i) limited scene diversity, (ii) limited caption diversity, (iii) single-instance annotations, and (iv) relatively small scale.

The proposed OS-W2S Label Engine aims to address these limitations and provide a data foundation for language-guided open-set aerial object detection. First, it leverages VLMs to expand existing aerial detection datasets with multi-granularity textual annotations (word-, phrase-, and sentence-level) based on six interpretable attributes. Second, its data preprocessing and postprocessing stages allow for handling of diverse aerial scenes and support multi-instance caption-box associations beyond the REC task constraint. Finally, by applying OS-W2S to eight widely used aerial detection datasets, we achieve 2 million image-caption pairs, which is 40× larger than those available in existing RS grounding datasets.

## C  OS-W2S Label Engine Details

**Predefined Size and Absolute Position Attributes.** In OS-W2S Label Engine, captions are generated based on six instance attributes: category, color, size, geometry, relative position, and absolute position. Among these attributes, the size (defined as the ratio of the instance's area to the image area) and absolute position (defined as the exact location of the instance within the image) are often subjectively determined. Moreover, the instances occupy only a very small portion of the image, which poses a challenge for the VLM to accurately determine the absolute positions of instances within the original images.

To address this issue, we apply predetermined rules to extract the size and absolute position attributes during the data pre-processing stage, explicitly providing this prior knowledge to the VLM during interaction. Specifically, we define size thresholds as [0.0005, 0.001, 0.01, 0.2], corresponding to bounding box area ratios relative to the image area, and categorize instances into ['tiny', 'small', 'medium', 'big', 'large']. Additionally, we segment the image into 25 regions using horizontal labels ['Far Left', 'Left', 'Center', 'Right', 'Far Right'] and vertical labels ['Top', 'Upper Middle', 'Middle', 'Lower Middle', 'Bottom'] to systematically define absolute position attributes.

**Foreground-Extraction Algorithm.** Aerial images typically contain numerous small, densely packed objects alongside complex backgrounds that occupy large portions of the image. This characteristic makes it difficult for a VLM to attend to the target instance when processing the raw image. Thus, a straightforward approach is to design a method that crops the foreground region for each instance, thereby effectively guiding the VLM's attention. However, excessive cropping may omit crucial contextual information, while insufficient cropping may result in missing important object details. To address this challenge, we design a foreground-extraction algorithm (Algorithm 1) that generates slightly expanded foreground crops based on bounding boxes. This enlarged crop includes the surroundings of the target object, while the target remains highlighted with a red box to guide the model in knowing exactly which object to describe. Providing this local contextual information enables the LVLM to accurately infer relative position attributes, maintaining a clear focus and reducing confusion caused by broader background noise.

---

**Algorithm 1** Foreground Region Extraction

---

**Require:** Bounding box set $B = \{b_1, b_2, \ldots, b_N\}$, image size $(w, h)$
**Ensure:** Foreground region set $R$
 1: **Step 1: Scale Bounding Boxes**
 2: **for** $i = 1$ to $N$ **do**
 3:     Compute the area $A_i$ of bounding box $b_i$
 4:     Determine scaling factor $s_i$ based on $A_i$
 5:     Update the bounding box $b_i$ to its extended version, ensuring it remains within the image boundaries
 6: **end for**
 7: **Step 2: Merge Overlapping Boxes**
 8: **for** each unmerged box $b_i$ **do**
 9:     Let $r \leftarrow b_i$
10:     **while** there exists an unmerged box $b_j$ that overlaps with $r$ **do**
11:         $r \leftarrow \text{MERGE}(r, b_j)$
12:         Mark $b_j$ as merged
13:     **end while**
14:     Add $r$ to the foreground region set $R$
15: **end for**
16: **return** $R$

---

**Matching Caption-Instance Pairs.** Existing remote sensing visual grounding datasets primarily focus on referring expression comprehension tasks, which are based on the assumption of associating one caption with a single instance. However, in real-world scenarios, a single caption often describes multiple instances that share similar attributes. Therefore, we aim to construct caption-instance pairs where one caption corresponds to multiple instances.

To achieve this goal, we design a caption-instance matching strategy (Algorithm 2) based on attribute similarity. Since our captions are composed of instance attribute combinations and each instance is annotated with corresponding attributes, we establish matching rules by comparing attribute similarity between captions and instances. This approach enables us to associate captions with all relevant instances that exhibit the described characteristics, thereby creating a more realistic and practical grounding dataset.

**Rationale for Selecting InternVL-2.5-38B-AWQ.** The quality of annotation is closely tied to the capability of the selected LVLM: larger models often deliver higher annotation quality but at greater computational cost.

---

**Algorithm 2** Caption-Instance Matching Strategy

---

1: **Input:**
2: - `captions`: A list of captions, each containing textual descriptions and associated attributes (e.g., category, size, color, geometry, relative position, absolute position).
3: - `instances`: A list of object instances, each identified by an ID and associated attributes (e.g., category, size, color, geometry, relative position, absolute position).
4: **Output:**
5: - `caption_instance_pairs`: A list of pairs $(caption, instance)$, where each caption is matched with corresponding object instances.
6: **Step 1: Initialization**
7: - Create an empty list `caption_instance_pairs` to store the matched caption-instance pairs.
8: **Step 2: Matching Process**
9: **for** each caption in `captions` **do**
10:      Extract relevant attributes (e.g., category, size, color) from the caption.
11:      Initialize an empty list `matched_instances` to store matching instances.
12:      **for** each instance in `instances` **do**
13:          Extract relevant attributes (e.g., category, size, color) from the instance.
14:          Compare attributes between caption and instance.
15:          **if** the attributes match sufficiently **then**
16:              Add instance to `matched_instances`.
17:          **end if**
18:      **end for**
19:      Add the pair $(caption, matched\_instances)$ to `caption_instance_pairs`.
20: **end for**
21: **Step 3: Output**
22: - Return `caption_instance_pairs`.

---

- **Initial model evaluation.** In a preliminary survey of leading LVLMs (e.g., Qwen2-VL and the InternVL2.5 family), InternVL2.5 emerged as the strongest open-source option at the time of dataset construction. Among all models tested, only *Qwen2-VL-72B*, *InternVL2.5-78B*, and *InternVL2.5-38B* (including quantized variants) consistently achieved a **100%** template-parsing success rate under regular-expression checks, indicating reliable adherence to our output schema.

- **Efficiency and cost analysis.** *InternVL2.5-78B* requires at least four 80 GB GPUs (4×A100), whereas *InternVL-2.5-38B-AWQ* runs on eight 24 GB GPUs (8×RTX 4090). In our trials, annotating 100 images took about 50 minutes with either configuration. Using typical rental prices (4×A100 $\approx$ \$4.18/h; 8×RTX 4090 $\approx$ \$2.09/h), the AWQ variant reduces hardware cost by roughly 50%, improving accessibility and reproducibility.

Balancing accuracy, efficiency, cost, and scalability, we finally select *InternVL-2.5-38B-AWQ* as the default annotator.

## D  MI-OAD DATASET DETAILS

**Base/Novel Categories Split.** To ensure that the MI-OAD dataset can strictly evaluate zero-shot transfer with novel classes, we split the categories into 75 base categories and 25 novel categories. The class division is based on clustering the semantic embeddings of the classes and selecting one class from each pair of leaf nodes in the clustering tree (Zang et al., 2024). The category splits are as follows:

- **Base:** 'aircraft', 'aircraft-hangar', 'airplane', 'baseball-diamond', 'baseball-field', 'bicycle', 'bridge', 'building', 'car', 'cargo-car', 'cargo-plane', 'cargo-truck', 'cement-mixer', 'chimney', 'construction-site', 'container', 'container-crane', 'container-ship', 'crane-truck', 'dam', 'damaged-building', 'dump-truck', 'engineering-vehicle', 'expressway-service-area', 'expressway-toll-station', 'facility', 'ferry', 'fishing-vessel', 'fixed-wing-aircraft', 'flat-car', 'front-loader-or-bulldozer', 'golf-field', 'ground-grader', 'harbor',

'haul-truck', 'helipad', 'hut-or-tent', 'large-vehicle', 'locomotive', 'oil-tanker', 'over-pass', 'passenger-car', 'passenger-vehicle', 'people', 'plane', 'pylon', 'railway-vehicle', 'roundabout', 'sailboat', 'shed', 'ship', 'shipping-container', 'small-aircraft', 'small-car', 'small-vehicle', 'soccer-ball-field', 'stadium', 'storage-tank', 'straddle-carrier', 'tank-car', 'tennis-court', 'tower', 'tower-crane', 'trailer', 'train-station', 'truck', 'truck-tractor', 'truck-tractor-with-flatbed-trailer', 'truck-tractor-with-liquid-tank', 'tugboat', 'utility-truck', 'van', 'vehicle', 'vehicle-lot', 'yacht'

- **Novel:** 'airport', 'awning-tricycle', 'barge', 'basketball-court', 'bus', 'crossroad', 'excavator', 'ground-track-field', 'helicopter', 'maritime-vessel', 'mobile-crane', 'motor', 'motor-boat', 'parking-lot', 'pedestrian', 'pickup-truck', 'playground', 'reach-stacker', 'scraper-or-tractor', 'shipping-container-lot', 'swimming-pool', 't-junction', 'tricycle', 'truck-tractor-with-box-trailer', 'windmill'

**MI-OAD Dataset Scale.** As shown in Table 4, we curated eight representative aerial detection datasets, and subsequently leveraged the OS-W2S Label Engine to enrich these datasets with multi-granularity textual annotations, which collectively constitute the MI-OAD dataset. Specifically, MI-OAD comprises 163,023 images and 2,389,973 (2M) image-caption pairs. As summarized in Table 5, MI-OAD is approximately 40 times larger than existing remote sensing grounding datasets and offers a robust data foundation for language-guided open-set aerial detection.

Table 4: Overview of the collected aerial-detection datasets. Image and instance counts are reported after cropping to a uniform resolution.

| Dataset | Images | Instances | Categories |
|---|---|---|---|
| DIOR (Li et al., 2020b) | 23,463 | 192,518 | 20 |
| DOTA v2.0 (Xia et al., 2018) | 19,871 | 495,754 | 18 |
| HRRSD (Zhang et al., 2019) | 44,002 | 96,387 | 13 |
| NWPU_VHR_10 (Su et al., 2019) | 1,244 | 6,778 | 10 |
| RSOD (Xiao et al., 2015) | 3,644 | 22,221 | 4 |
| SODA-A (Pisani et al., 2024) | 31,798 | 1,008,346 | 9 |
| VisDrone (Zhu et al., 2021) | 29,040 | 740,419 | 10 |
| xView (Lam et al., 2018) | 9,961 | 732,960 | 60 |

Table 5: Comparison with existing remote-sensing grounding datasets.

| Dataset | Categories | Images | Image–Caption Pairs |
|---|---|---|---|
| RSVG-H (Sun et al., 2022) | – | 4,239 | 7,933 |
| DIOR-RSVG (Zhan et al., 2023) | 20 | 17,402 | 38,320 |
| OPT-RSVG (Li et al., 2024) | 14 | 25,452 | 48,952 |
| MI-OAD (ours) | 100 | 163,023 | 2M |

# E  MORE EXPERIMENTAL RESULTS

## E.1  TRAINING DETAILS.

### E.1.1  BASELINES AND EXPERIMENTAL SETUP.

To demonstrate the effectiveness of MI-OAD, we conduct experiments on three representative tasks: (i) language-guided open-set aerial object detection; (ii) remote-sensing visual grounding (RSVG); (iii) open-vocabulary aerial detection (OVAD).

For language-guided open-set aerial object detection, we evaluate two representative language-guided open-set detectors, Grounding DINO (Liu et al., 2024) and YOLO-World (Cheng et al., 2024), on MI-OAD at three semantic granularities: *vocabulary*, *phrase*, and *sentence*. This constitutes the first comprehensive benchmark for language-guided open-set aerial object detection. We adopt the MMDetection implementation of Grounding DINO and the official v1.0 release of YOLO-World. Unless otherwise specified, all experiments are executed on 32 NVIDIA RTX 4090 GPUs

with a batch size of four per GPU. Grounding DINO is trained for 12 epochs, whereas YOLO-World is trained for 40 epochs; all other hyper-parameters remain at their default values.

For remote-sensing visual grounding, we use Grounding DINO as the baseline and evaluate it on two standard benchmarks: DIOR-RSVG and OPT-RSVG. We first report Grounding DINO performance when fine-tuning on each training set, and subsequently examine the model's performance when adding MI-OAD datasets. All fine-tuning experiments are conducted on eight NVIDIA RTX 4090 GPUs for 12 epochs with a batch size of four per GPU, while keeping all other hyper-parameters at their default settings. During evaluation, since RSVG focuses on the Referring Expression Comprehension (REC) task, we retain only the bounding box with the highest confidence score and report standard metrics (Pr@{0.5, 0.6, 0.7, 0.8, 0.9}, mean IoU, and cumulative IoU). We also compare the resulting scores with those of current state-of-the-art methods, including MGVLF (Zhan et al., 2023) and LPVA (Li et al., 2024), to quantify the gains afforded by MI-OAD pre-training.

For open-vocabulary aerial detection, we incorporate the SOTA method LAE-DINO and retrain it on LAE-1M with 32 RTX-4090 GPUs using default hyperparameters for fair comparison. We then conduct additional training with MI-OAD to evaluate the performance improvements.

### E.1.2 Task Formulation.

Most existing language-guided open-set detectors (Liu et al., 2024; Cheng et al., 2024) focus on language-guided detection and phrase grounding, where the goal is to identify and localize all nouns mentioned in descriptions. In contrast, our dataset targets detection and generalized referring expression comprehension (GREC), where GREC requires finding one or multiple instances of the same category that satisfy a descriptive caption. To seamlessly integrate with existing frameworks, we unify the GREC task as a detection paradigm. Specifically, since each caption in our dataset corresponds to a unique category label, we can simply replace detection class names with their corresponding captions during training.

### E.1.3 Prompt Construction Strategy

Prompt construction plays a crucial role in both training and inference phases. To enhance model robustness, we apply a randomized category sampling strategy during training. Specifically, for each detection sample, we define categories present in the image as positive classes ($C_{pos}$) and consider the remaining categories as negative classes ($C_{neg}$). We include all positive classes and randomly select between 1 and $|C_{neg}|$ negative classes to form the textual prompt associated with each sample.

However, since MI-OAD integrates eight distinct detection datasets, category conflicts across datasets may arise. For instance, an image containing an object labeled as *airplane* should consider *airplane* as a positive class; however, related categories such as *aircraft* could incorrectly appear among negative classes. To prevent such conflicts, we restrict negative class sampling strictly to categories from the same original dataset, as the annotations within each source dataset are manually verified and thus do not contain conflicting category labels. For grounding samples, we adopt a consistent approach by simply replacing the positive class labels with the corresponding image captions.

During inference, detection samples utilize prompts consisting of all categories from their respective source datasets, while grounding samples use prompts composed solely of their corresponding captions.

### E.2 Impact of Dataset Scale

To investigate whether existing datasets can support language-guided open-set aerial detection, we conduct a comparative analysis using the largest available RSVG dataset (OPT-RSVG, ∼0.05M) and open-vocabulary detection dataset (LAE-1M, ∼0.18M) against our MI-OAD (∼2M). As shown in Table 6, this comparison reveals a critical gap in current resources.

Models pre-trained on existing datasets struggle significantly with language-guided open-set tasks. Grounding DINO achieves only 3.3% detection $AP_{50}$ and 7.0% sentence-level $AP_{50}$ when trained on OPT-RSVG. LAE-DINO performs better with LAE-1M pre-training (24.8% detection $AP_{50}$) but still falls short on complex grounding tasks, achieving merely 5.1% for sentence-level grounding.

Table 6: Impact of pre-training data scale on language-guided open-set aerial detection performance. Models pre-trained on existing datasets (OPT-RSVG, LAE-1M) versus MI-OAD are evaluated on the MI-OAD Val-FT set across detection and grounding tasks.

| Method | Pre-Training Data | Detection | | Phrase | | | | Sentence | | | |
|---|---|---|---|---|---|---|---|---|---|---|---|
| | | $AP_{50}$ | R@100 | $AP_{50}$ | R@1 | R@10 | R@100 | $AP_{50}$ | R@1 | R@10 | R@100 |
| Grounding DINO | OPT-RSVG | 3.3 | 35.4 | 9.7 | 15.0 | 33.3 | 34.2 | 7.0 | 14.5 | 26.1 | 26.4 |
| **Grounding DINO** | **MI-OAD** | **37.1** | **70.1** | **57.8** | **35.2** | **74.4** | **91.5** | **56.4** | **44.1** | **78.0** | **90.3** |
| Gain over Grounding DINO | | +33.8 | +34.7 | +48.1 | +20.2 | +41.1 | +57.3 | +49.4 | +29.6 | +51.9 | +63.9 |
| LAE-DINO | LAE-1M | 24.8 | 70.1 | 22.9 | 22.9 | 56.0 | 79.7 | 5.1 | 17.5 | 47.1 | 67.9 |
| **LAE-DINO** | **MI-OAD** | **40.0** | **75.0** | **59.4** | **35.8** | **74.9** | **92.6** | **57.8** | **44.5** | **78.9** | **92.4** |
| Gain over LAE-DINO | | +15.2 | +4.9 | +36.5 | +12.9 | +18.9 | +12.9 | +52.7 | +27.0 | +31.8 | +24.5 |

In contrast, pre-training on MI-OAD yields consistent and substantial improvements. Grounding DINO's detection $AP_{50}$ increases to 37.1% (11× improvement), while sentence-level grounding $AP_{50}$ improves to 56.4% (8× improvement). Similarly, LAE-DINO shows dramatic gains, with sentence-level $AP_{50}$ rising from 5.1% to 57.8%.

These substantial improvements underscore a fundamental limitation: existing aerial datasets lack the scale and diversity required for effective language-guided open-set detection. By providing 40× more grounding annotations than previous datasets, MI-OAD establishes the critical data foundation necessary for advancing language-guided open-set aerial detection tasks.

## F   LIMITATION ANALYSIS

While the OS-W2S Label Engine and MI-OAD advance aerial object detection research, our approach has inherent trade-offs. We prioritized annotation quality by building MI-OAD upon eight well-established datasets with human-verified annotations, ensuring a trustworthy benchmark for this first language-guided open-set aerial detection dataset. However, this design choice inevitably limits geographic, temporal, and environmental diversity. Despite our efforts to maximize scene variety through dataset integration, certain regions, seasons, and conditions remain underrepresented. Future work could address this limitation by incorporating more diverse sources like OpenStreetMap and Google Earth Engine.

## G   QUALITATIVE ANALYSIS OF LANGUAGE-GUIDED OPEN-SET AERIAL DETECTION RESULTS

In this section, we demonstrate and analyze the effectiveness of our proposed dataset from three perspectives. First, we compare and visualize the detection results of Grounding DINO with and without domain adaptation using MI-OAD. Second, to simulate realistic application scenarios, we evaluate the model's language-guided open-set aerial detection capability trained on the MI-OAD dataset using manually defined prompts that are not included in the dataset annotations. Finally, we visualize the model's performance on the MI-OAD Validation Set by employing prompts at three granularity levels: vocabulary-level, phrase-level, and sentence-level.

### G.1   COMPARISON OF GROUNDING DINO WITH AND WITHOUT DOMAIN ADAPTATION

Fig. 4 visualizes the detection results of Grounding DINO before and after domain adaptation training on our MI-OAD P-Set dataset. As observed in the results of the first and second columns, Grounding DINO, originally designed for natural images, exhibits a considerable domain gap when directly applied to aerial imagery domains. However, after domain adaptation using our proposed dataset, the detection results significantly improve. From the third column, we observe that while Grounding DINO can localize objects in common urban scenarios, it exhibits clear false positives and missed detections—for example, incorrectly detecting a green taxi with the prompt "white car" and missing smaller white cars in the distance. Following training on our dataset, the model notably improves its ability to detect smaller instances and accurately recognize instance attributes.

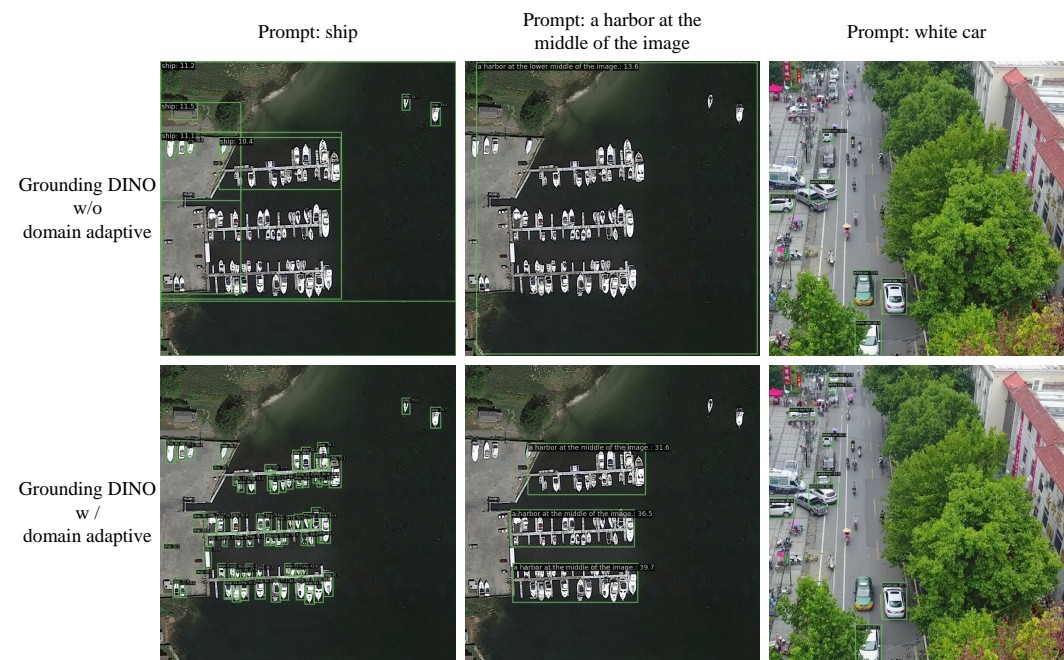

Figure 4: Visualization of detection results comparing GroundingDINO without and with domain adaptation using our proposed MI-OAD P-Set dataset.

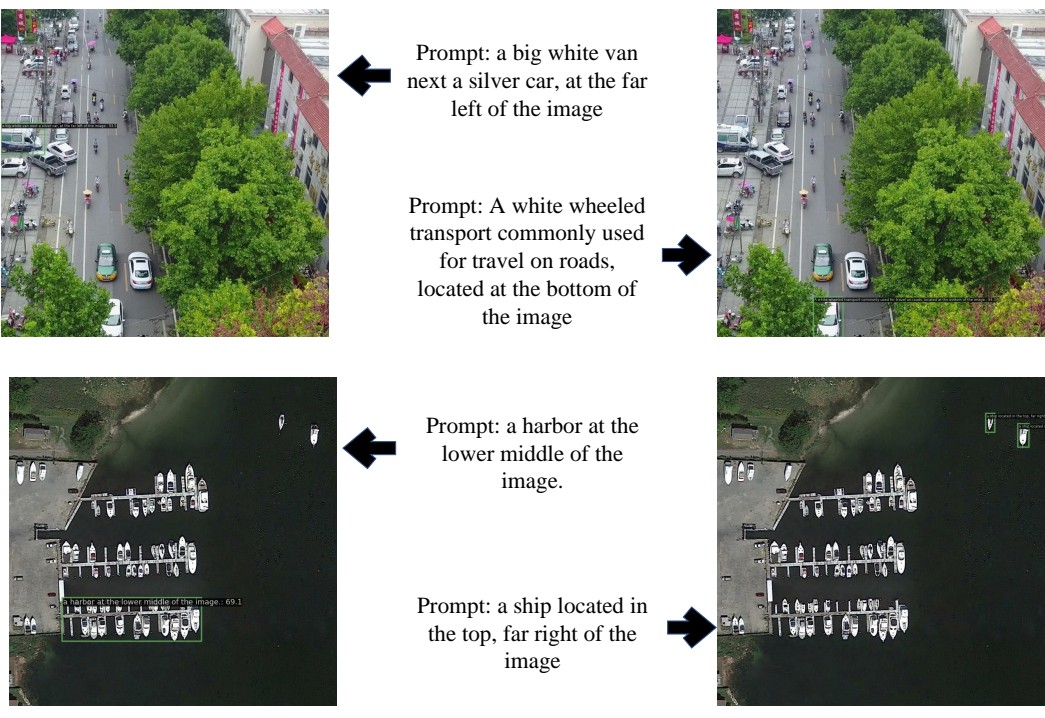

Figure 5: Qualitative visualization of language-guided open-set aerial detection performance with self-defined prompts (Part 1)

Prompt: a green taxi on the street
Prompt: a white car at the bottom of the image
Prompt: a white car is to the right of the green taxi

Prompt: a yellow motor on the road
Prompt: a people wearing a yellow hat
Prompt: a van near the utility pole

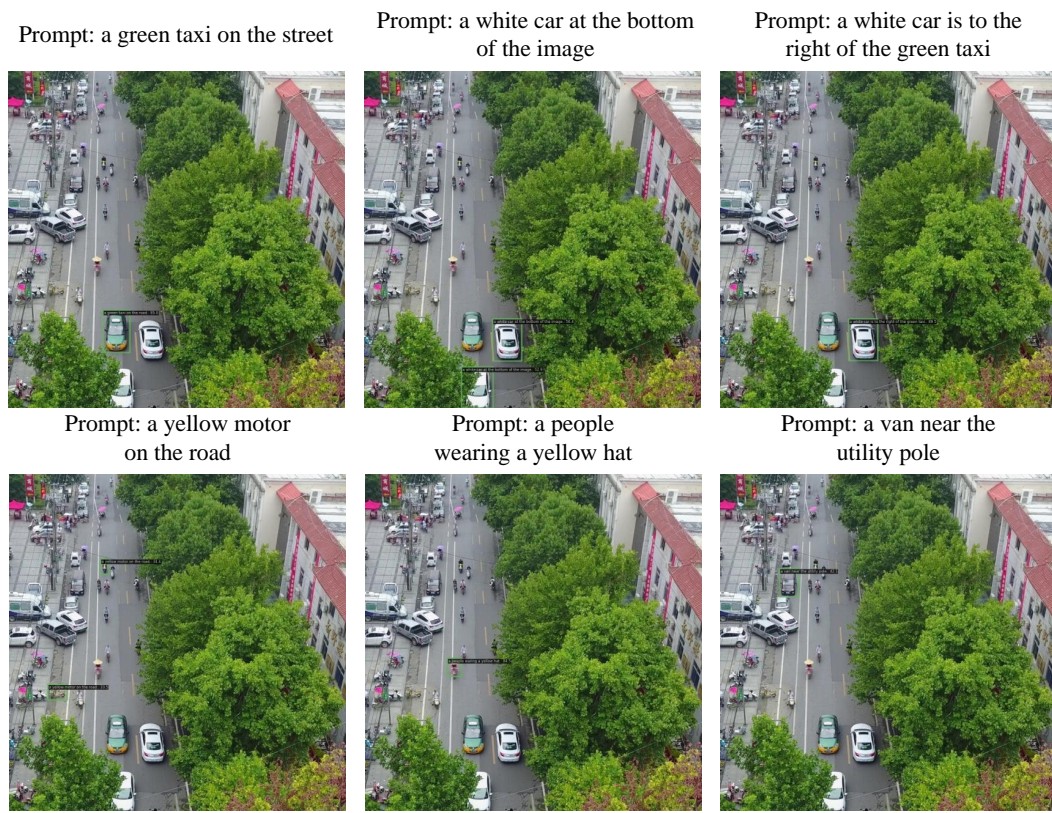

Figure 6: Qualitative visualization of language-guided open-set aerial detection performance with self-defined prompts (Part 2)

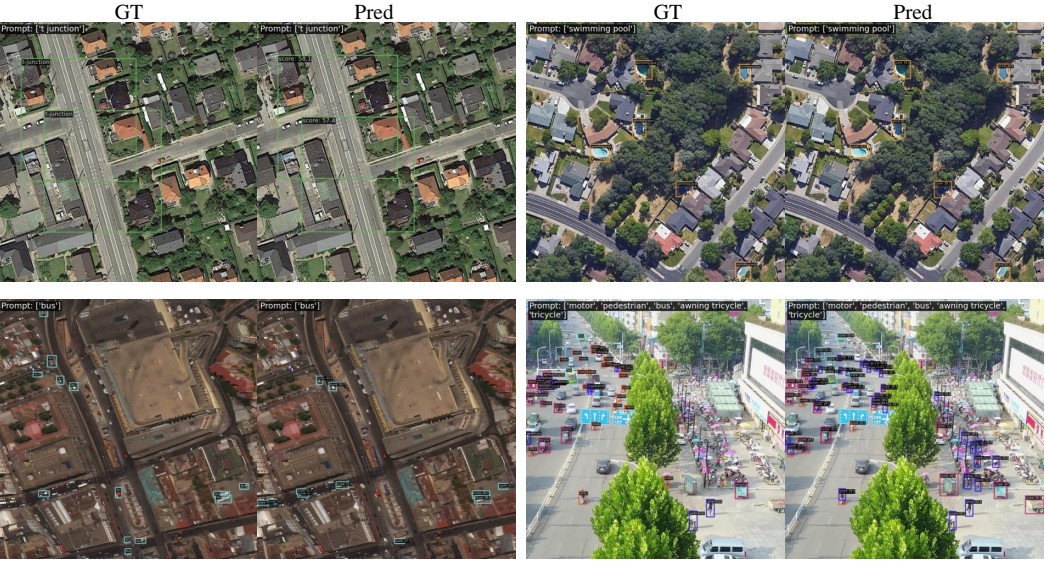

Figure 7: Visualization of language-guided open-set aerial detection results at the vocabulary-level using annotation-derived prompts.

## G.2 EVALUATION OF LANGUAGE-GUIDED OPEN-SET AERIAL DETECTION USING MANUALLY DEFINED PROMPTS

To further demonstrate the practical efficacy of our dataset, we simulate realistic application scenarios by employing manually defined prompts that are not included in the dataset annotations to

evaluate the language-guided open-set aerial detection capability of Grounding DINO after domain adaptation using our proposed dataset.

Fig. 5 visualizes detection results from the model in urban and harbor scenarios. Notably, in the second column of the first row, the model successfully detects objects described by implicitly defined prompts, where object categories are not explicitly mentioned but are described solely through attributes. This capability can be attributed to the attribute-based captions in our dataset and Grounding DINO's sentence-level image-text alignment approach. Additional examples also highlight the model's sensitivity to relative and absolute positional information.

As illustrated in Fig. 6, we further evaluate the model's language-guided open-set aerial detection performance using prompts at different granularity levels, ranging from vocabularies to phrases, and ultimately to sentences. To intuitively demonstrate the influence of different prompt complexities on model performance, we conduct tests on the same image. The first column in the first row demonstrates the model's strong generalization capability, accurately detecting objects corresponding to prompts including novel classes, such as "a green taxi on the street." Moreover, the model shows strong sensitivity to relative positional attributes, as highlighted in the third column of the first row, where two white cars near a green taxi are accurately identified based on the prompt "right of the green taxi," further indicating the model's spatial understanding capability. Additionally, as shown in the third row, the model achieves highly precise detection results for small targets.

### G.3 MULTI-GRANULARITY LANGUAGE-GUIDED DETECTION VISUALIZATION

We also visualize the model's language-guided open-set detection capability on the MI-OAD Validation Set using prompts derived from annotations. Specifically, we illustrate the detection performance at three different prompt complexity levels: vocabulary-level, as shown in Fig.7; phrase-level, as shown in Fig.8; and sentence-level, as shown in Fig. 9.

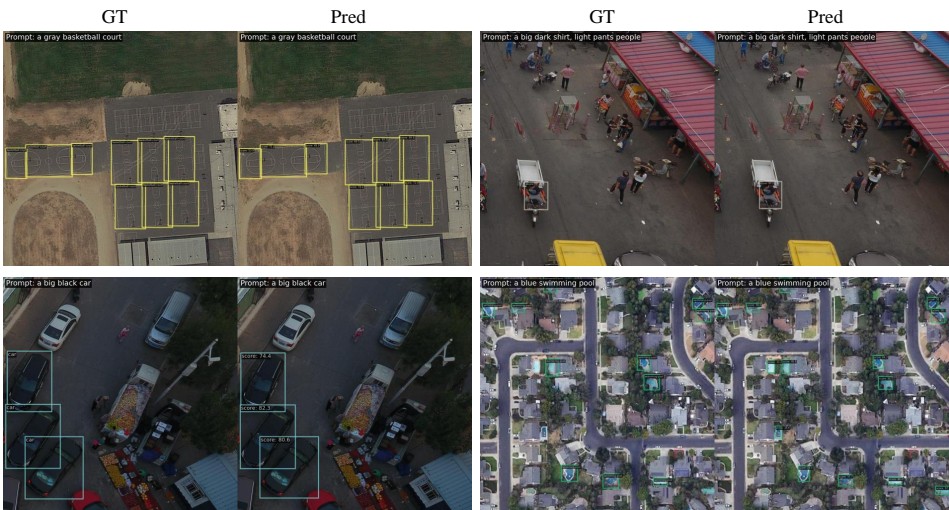

Figure 8: Visualization of language-guided open-set aerial detection results at the phrase-level using annotation-derived prompts.

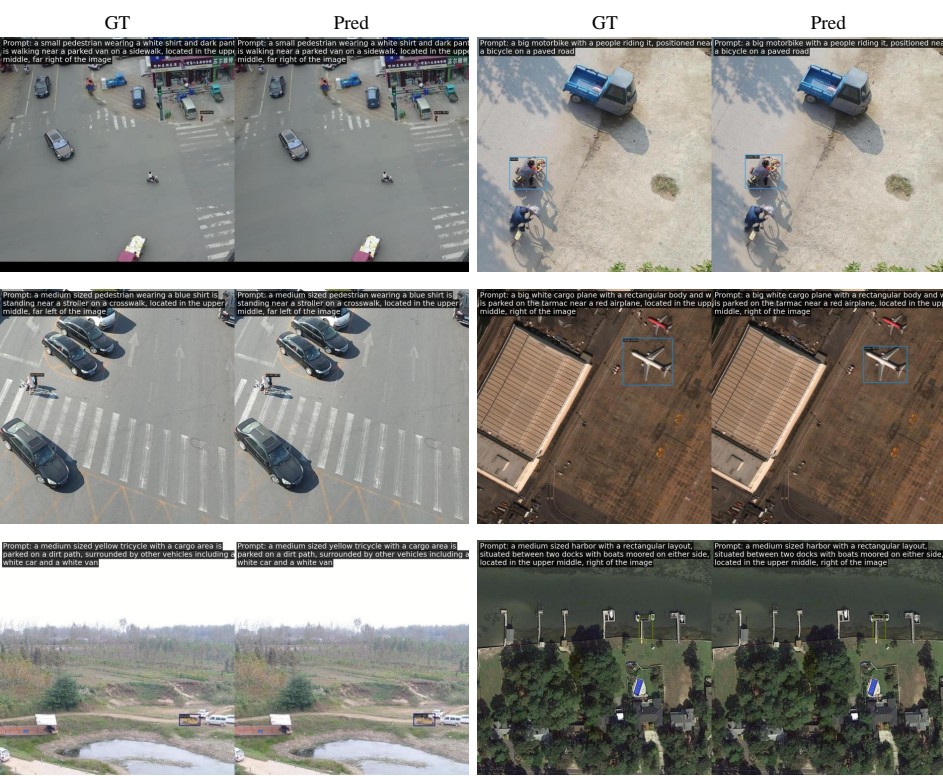

Figure 9: Visualization of language-guided open-set aerial detection results at the sentence-level using annotation-derived prompts.

## H  LABEL ENGINE PROMPTS

---

**Introduction**

Hello InterVL,
I need your assistance in annotating aerial images. We will proceed in three steps:

1. **Initial Captioning**: I will provide an image of an aerial target. Please generate a caption describing its attributes including {gt_size} and {gt_category}.

2. **Caption Refinement with Context**: Next, I'll provide an image showing the target within its surroundings. Please refine the caption by adding information about the target's relative location within its environment.

3. **Caption Enhancement with Absolute Location**: Finally, I'll provide the region of the image where the target is located (for example: top, left of the image). Based on this information and the caption from Step 2, please incorporate the absolute location attribute.

**Important**: The red box in the provided images is only for your reference to identify the target. **Do not** mention the red box or any red-box-related information in the final caption.

---

**Output Templates**

```
caption1_template = {
    • "caption":  f"[A brief sentence describing the target
      using the provided Category and Size.  Include **Color**
      and **Geometry** only if you are certain about them.]",
    • "Category":  f"{{gt\_category}}",
    • "Size":  f"{{gt\_size}}",
    • "Color":  "[Include if certain]",
    • "Geometry":  "[Include if certain]"
}
caption2_template = {
    • "caption":  f"[Refined caption including the target's
      relative location attribute.]",
    • "relative_location":  "[The target's relative location
      within its surroundings.]"
}
caption3_template = {
    • "caption":  "[The caption by incorporating the absolute
      location.]",
    • "absolute_location":  f"{{box\_pos}}"
}
```

---

**Prompts (R1–R3)**

**R1 — Step 1 (Category, Size, Geometry, Color)**
```
<Instance region image>
```
You are provided with an aerial image of a target. The red box highlights the target.

- Generate a caption describing the target.
- **Must** using the provided **Category:** "{gt_category}" and **Size:** "{gt_size}" in caption.

---

- Include **Color** and **Geometry** only if you are certain about them.

- Do **not** mention the red box or any red box-related information in final caption.

- Keep the caption under 20 words.

- Only include information you can confidently determine from the image. Avoid speculative or aesthetic descriptions.

**Must format your answer as a JSON object with the following structure and strictly adhere to the JSON format:** {caption1_template}

**R2 — Step 2 (relative location)**
<Instance foreground image>
You are provided with an instance's foreground region image showing its surrounding environment. The red box highlights the target (for your reference, do not mention it).

- Based on the caption from Step 1: "{self_caption}", refine the description by incorporating **relative location** information about the target with respect to its surrounding environment or nearby objects.

- Maintain the original attributes (**Category**, **Size**, **Color**, **Geometry**).

- Do **not** mention the red box or any red box-related information in final caption.

- Do **not** describe the target's location relative to the image boundaries (e.g., 'top left of the image').

- Keep the caption under 40 words.

- Only include information you can confidently determine from the image. Avoid speculative or aesthetic descriptions.

**Must format your answer as a JSON object with the following structure and strictly adhere to the JSON format:** {caption2_template}

**R3 — Step 3 (absolute location)**
You are provided the instance's absolute position in the image.
**Absolute Location**: "{box_pos}".

- Review the caption from Step 2: "{relative_caption}", enhance the caption by incorporating the provided **absolute location** information.

- Keep the caption under 60 words.

- Only include information you can confidently determine from the image. Avoid speculative or aesthetic descriptions.

**Must format your answer as a JSON object with the following structure and strictly adhere to the JSON format:** {caption3_template}

