# OpenReview forum: "OS-W2S: An Automatic Labeling Engine for Language-Guided Open-Set Aerial Object Detection"
_ICLR.cc/2026/Conference — Submitted to ICLR 2026_

### Official Review · Reviewer_zebP · 2025-10-29

**Soundness:** 3
**Presentation:** 3
**Contribution:** 3
**Rating:** 4
**Confidence:** 4

**Summary:**

This paper presents OS-W2S Label Engine, an automatic annotation pipeline capable of handling diverse scene annotations for aerial images. Using this label engine, the authors also construct a benchmark dataset MI-OAD for language-guided open-set aerial detection. Experiments on open-vocabulary aerial detection and remote sensing visual grounding are carried by training and evaluating the model with MI-OAD.

**Strengths:**

1. MI-OAD contains 163K images and 2 million image-caption pairs, with multiple instances per caption, which is large-scale compared to existing datasets.
2. The experiments show that using the proposed MI-OAD as training corpus results in a better performance in both open-vocabulary detection and visual grounding tasks.

**Weaknesses:**

1. While MI-OAD contains 163,023 images and 2 million image-caption pairs, it is difficult to evaluate a dataset simply based on its size. For example, if we split each image into four, then the number will be 4x larger. And for MI-OAD, its quality is not guaranteed. The captions are generated by VLMs, but there is not enough experiments involving quality assessment.
2. The dataset is essentially generated by integrating available datasets. The problem is, after reviewing some of the data, I find that most of them keep the original categories, without unification. From what I understand, most of the data are labeled with only a small portion of the 100 categories mentioned in the paper, and the other objects are just ignored.
3. Some minor writing issues: For example, in Page 6, Fig.3b, Section5.1: Missing space. ( 1,765 instances): Redundant space. In Fig. 2, InterVL should be InternVL. In Fig. 3, (e) is not aligned, and some labels in (a)-(d) are too small.

**Questions:**

1. Images in RS show high variation. Some of them contain only a few objects, some may contain more than 100 objects. How do you control the captions in these two different situations? If there are many objects in the image, what will the caption be like?
2. In figure 3, "instances per caption" shows 98% images have <= 20 instances. But many datasets used by the authors (such as DOTA) are originally very dense. Why is this discrepancy? Do you crop the image to decrease the instance count? If yes, is the preprocessing rule detailed in the paper?

---

> ### Author Response · Authors · 2025-11-21
> **Response to Weaknesses 1 (1/2)**
>
> Thank you for your valuable time and constructive feedback. We hope to address your main concerns below and will revise the paper accordingly if it is accepted. Looking forward to your further discussion!
>
> ---
>
> > Weaknesses comment 1. While MI-OAD contains 163,023 images and 2 million image-caption pairs, it is difficult to evaluate a dataset simply based on its size. For example, if we split each image into four, then the number will be 4x larger.
>
> Thank you for pointing this out.
>
> **Rationale for Reporting 163,023 Images**
>
> This count is designed to reflect two aspects:
>
> 1. **Scene diversity**: These images come from eight diverse aerial detection datasets captured at different altitudes and viewpoints, ensuring varied real-world scenarios.
>
> 2. **Standard preprocessing for aerial imagery**: Due to the high resolution of aerial images, cropping large images into smaller patches (typically 800×800 or 1024×1024 pixels) is standard practice in most aerial detection work. The 163,023 count reflects these non-duplicate cropped images used across training, validation, and test sets, representing the practical dataset scale.
>
> **Rationale for Reporting 2 Million Image-Caption Pairs**
>
> Following established conventions in grounding tasks, we report image-caption pairs as the primary scale metric because each caption paired with its image forms an independent grounding sample. This yields 2 million samples, which is directly comparable to prior work: DIOR-RSVG has 38,320 pairs and OPT-RSVG has 48,952 pairs. This metric accurately reflects the annotation effort and dataset scale.
>
> **Planned Revision**
>
> Following your suggestion, we will update Appendix Table 4 to include the original image counts before cropping for each source dataset. This will show both the number of unique high-resolution scenes and the preprocessed image count (163,023), providing complete transparency.
>
> **Updated Table 4:**
>
> | Dataset     | Original Images | Preprocessed Images | Instances     | Categories |
> | ----------- | --------------- | ------------------- | ------------- | ---------- |
> | DIOR        | 23,463          | 23,463              | 192,518       | 20         |
> | DOTA v2.0   | 2,423           | 19,871              | 495,754       | 18         |
> | HRRSD       | 21,761          | 44,002              | 96,387        | 13         |
> | NWPU VHR-10 | 650             | 1,244               | 6,778         | 10         |
> | RSOD        | 976             | 3,644               | 22,221        | 4          |
> | SODA-A      | 2,513           | 31,798              | 1,008,346     | 9          |
> | VisDrone    | 10,209          | 29,040              | 740,419       | 10         |
> | xView       | 846             | 9,961               | 732,960       | 60         |
> | **Total**   | **62,841**      | **163,023**         | **3,295,383** | **100**    |
>
> **Note:** "Original Images" refers to the number of images in the source datasets before resolution standardization. "Preprocessed Images" reflects the images after cropping to uniform resolution across all splits. For xView, we use only the annotated training set. For DOTA v2.0, we use the train and val sets. For NWPU VHR-10, we include only the positive samples with annotations.

---

> > ### Author Response · Authors · 2025-11-21
> > **Response to Weaknesses 1 (2/2)**
> >
> > ### Response to Weaknesses 1 (2/2)
> >
> > > Weaknesses comment 1. And for MI-OAD, its quality is not guaranteed. The captions are generated by VLMs, but there is not enough experiments involving quality assessment.
> >
> > Thank you for raising this point. As the reviewer correctly notes, ensuring data quality is critical for dataset construction. We carefully considered this throughout our design process and implemented multiple safeguards. Ultimately, the substantial improvements on existing benchmarks (Tables 2 and 3) provide indirect evidence of our dataset quality. Below, we detail the measures taken to ensure annotation quality and the experimental results validating the dataset quality.
> >
> > ---
> >
> > ### **High-Quality Source Datasets**
> >
> > We construct MI-OAD based on eight widely-used aerial object detection datasets, including VisDrone, DOTA, and DIOR, among others. These datasets have become standardized benchmarks within the aerial detection community due to their high quality, diversity, and representativeness across various sensor platforms (e.g., UAVs, satellites) and image resolutions. This careful selection ensures that our foundation is built on verified, human-annotated detection labels.
> >
> > ### **Minimizing Domain Gap for VLM Annotation**
> >
> > To address the domain gap between VLMs trained on natural images and aerial imagery, we implement two key strategies:
> >
> > **1. Ensuring Attribute Correctness through Reliable Priors**
> >
> > The quality of our annotations depends on the accuracy of six instance attributes: category, size, color, geometry, relative position, and absolute position. To reduce errors from domain gap, we explicitly provide the VLM with three prior attributes per instance:
> >
> > - **Category**: Deterministically obtained from object detection annotations.
> > - **Size**: Computed as the area ratio using bounding box dimensions.
> > - **Absolute Location**: Determined by the instance's coordinates within the image using rule-based methods.
> >
> > Providing these prior attributes reduces ambiguity and helps the VLM generate more reliable descriptions. The remaining three attributes (color, geometry, relative position) are generated by the VLM based on visual content, as they are relatively robust to domain shifts and can be accurately inferred from properly cropped regions.
> >
> > **2. Optimizing Visual Input through Two-Stage Cropping**
> >
> > Appropriate cropping helps the VLM focus on the target while providing sufficient context. Through iterative experimentation, we developed a two-stage strategy:
> >
> > - **Instance Region Cropping**: We crop each object based on its bounding box and enlarge it by a scale factor (Figure 2). Combined with category and size priors, this ensures the VLM concentrates on the specific instance, enabling accurate generation of color and geometry attributes while minimizing misclassifications.
> >
> > - **Foreground Region Extraction**: We generate a slightly expanded foreground crop that includes the immediate surroundings of the target object, with the target highlighted by a red box (as shown in Figure 2). This provides essential local context for inferring relative position attributes while reducing confusion from broader background noise.
> >
> > ### **Quality Validation**
> >
> > We ensure annotation quality through multiple validation approaches:
> >
> > **1. Structured Output Validation**
> >
> > All VLM outputs follow a predefined JSON format validated through regular expression parsing. This ensures 100% parsing success and structural consistency across all annotations.
> >
> > **2. Independent VLM-Based Quality Assessment**
> >
> > We sampled 300 images (approximately 1,765 instances) from MI-OAD and validated the VLM-generated attributes using two independent, powerful models: InternVL3-78B (a stronger open-source model) and GPT-4o-mini (a leading closed-source model). The results demonstrate consistently high accuracy:
> >
> > - **InternVL3-78B**: color (98.98%), geometry (99.21%), relative position (97.90%)
> > - **GPT-4o-mini**: color (96.88%), geometry (94.39%), relative position (96.92%)
> >
> > These high accuracy rates provide quantitative evidence that our domain gap mitigation strategies are effective.
> >
> > **3. Downstream Task Performance**
> >
> > MI-OAD's effectiveness is further validated through substantial improvements on multiple existing benchmarks:
> >
> > - **Remote Sensing Visual Grounding (Table 2)**: Achieves state-of-the-art results on both DIOR-RSVG and OPT-RSVG datasets, with significant gains in high-precision metrics (e.g., +12.52 points in Pr@0.9 on OPT-RSVG).
> > - **Open-Vocabulary Aerial Detection (Table 3)**: Notable performance gains across DIOR (+7.1 AP50), DOTA-v2.0 (+5.2 mAP), and LAE-80C (+2.5 mAP) benchmarks.
> >
> > These consistent improvements across diverse tasks demonstrate that MI-OAD provides high-quality annotations that enable effective model training.
> >
> > **4. Manual Verification of Test Set**
> >
> > To establish MI-OAD as a reliable benchmark, we conducted rigorous manual reviews during test set construction.
> >
> > ---

---

> ### Author Response · Authors · 2025-11-21
> **Response to Weaknesses 2**
>
> > Weaknesses comment 2. The dataset is essentially generated by integrating available datasets. The problem is, after reviewing some of the data, I find that most of them keep the original categories, without unification. From what I understand, most of the data are labeled with only a small portion of the 100 categories mentioned in the paper, and the other objects are just ignored.
>
> ---
>
> Thank you for this insightful observation. We fully understand your concern regarding the lack of category unification and the absence of pseudo-labeling across all 100 categories. We explain our design rationale below and acknowledge valuable directions for future work.
>
> **Rationale for Preserving Original Categories Without Unification**
>
> We intentionally preserved the original category labels from source datasets without merging synonyms. This decision was motivated by the nature of open-vocabulary detection: aligning visual features of the same object with different textual expressions (e.g., "car" vs. "vehicle") helps strengthen the model's open-vocabulary detection capability by encouraging robust image-text alignment across diverse linguistic expressions.
>
> To avoid conflicts during training where multiple synonym categories might be sampled simultaneously while only one has corresponding annotations (e.g., the prompt contains both "car" and "vehicle," but the object is only annotated as "car"), we implemented a careful negative sampling strategy (detailed in Appendix E.1.3). Specifically, when constructing prompts for detection samples, we restrict negative class sampling to categories from the same source dataset, ensuring no synonym conflicts occur.
>
> **Rationale for Not Adding Pseudo-Labels Across All 100 Categories**
>
> As you correctly observed, we did not generate pseudo-labels to annotate all objects according to the unified 100 categories. This decision prioritizes annotation quality: we selected eight datasets specifically for their rigorous human annotation processes and high-quality bounding boxes. Since bounding box accuracy is critical for detection tasks, we avoided introducing pseudo-labels that could compromise the precision of the original annotations.
>
> By preserving the original annotations without modification, we maintain both transparency and flexibility. We have preserved comprehensive metadata from the source datasets, including dataset names and image identifiers, allowing researchers to trace, filter, or customize the dataset according to their specific requirements.
>
> **Future Directions**
>
> We strongly agree that category unification and comprehensive pseudo-labeling are valuable directions worth pursuing. We plan to address these in MI-OAD v2 by:
>
> 1. **Building a category taxonomy** using VLMs to unify the category vocabulary across datasets.
> 2. **Exploring reliable pseudo-labeling methods** to expand annotations for existing images and annotate previously unlabeled objects, thereby increasing dataset scale.
>
> ---

---

> ### Author Response · Authors · 2025-11-21
> **Response to Weaknesses 3**
>
> > Weaknesses comment 3. Some minor writing issues: For example, in Page 6, Fig.3b, Section5.1: Missing space. ( 1,765 instances): Redundant space. In Fig. 2, InterVL should be InternVL. In Fig. 3, (e) is not aligned, and some labels in (a)-(d) are too small.
>
> Thank you for your careful review. We have corrected all the problems you mentioned in the revised manuscript. We will conduct a thorough review of the entire manuscript to ensure no similar issues remain.

---

> ### Author Response · Authors · 2025-11-21
> **Response to Question 1**
>
> > Question comment 1. Images in RS show high variation. Some of them contain only a few objects, some may contain more than 100 objects. How do you control the captions in these two different situations? If there are many objects in the image, what will the caption be like?
>
> ---
>
> Thank you for this important question. The OS-W2S Label Engine is designed to handle aerial scenes with varying object densities through an **instance-level annotation approach**. The key advantage is that caption generation for each instance remains independent of the total number of objects in the image.
>
> Our pipeline operates at the instance level through three stages:
>
> **1. Data Preprocessing (per instance)**
>
> As discussed in our response to Comment 2/5, for each individual instance in the image, we extract:
>
> - **Reliable attribute priors** (category, size, absolute location) from detection annotations using rule-based methods.
> - **Instance region crops** based on bounding boxes to focus the VLM on specific targets.
> - **Foreground region crops** to provide local context for inferring relative position attributes while reducing confusion from broader background noise.
>
> This preprocessing is performed independently for each instance, regardless of how many objects exist in the image.
>
> **2. Instance-Level Caption Generation (per instance)**
>
> Using the preprocessed inputs, the VLM generates attributes and captions for each instance independently (as shown in Figure 2). Each object receives:
>
> - Six attributes: category, size, color, geometry, relative position, and absolute position.
> - Three sentence-level captions with varying detail levels.
> - Three phrase-level captions based on attribute combinations.
>
> **3. Data Post-Processing (establishing caption-instance associations)**
>
> Since multiple instances may share similar attributes, we establish caption-instance associations by computing compositional attribute similarity between captions and instances. This enables a single caption to match multiple instances when appropriate (detailed in Appendix C).
>
> ---

---

> ### Author Response · Authors · 2025-11-21
> **Response to Question 2**
>
> > Question comment 2. In figure 3, "instances per caption" shows 98% images have <= 20 instances. But many datasets used by the authors (such as DOTA) are originally very dense. Why is this discrepancy? Do you crop the image to decrease the instance count? If yes, is the preprocessing rule detailed in the paper?
>
> ---
>
> Thank you for pointing this out. We clarify below:
>
> **(1) Figure 3(d) shows instances per caption, not instances per image**
>
> Figure 3(d) presents the distribution of **instances per caption**, which reflects how many objects match each caption based on the caption's specificity. This is different from the number of objects in an image.
>
> The number of instances matched by a caption depends on the caption's level of detail:
>
> - **Highly specific captions** (e.g., "a big ship with a rectangular cargo hold is docked next to another similar ship, located at the center of the image") typically match only 1-2 instances due to their detailed attribute combinations and spatial constraints.
> - **Generic captions** (e.g., "a big ship" or "a gray ship") can match more instances sharing the same attributes.
>
> Since MI-OAD generates captions with varying levels of detail and attribute combinations, most captions are sufficiently specific to match a limited number of instances. This explains the distribution shown in Figure 3(d).
>
> **(2) Image cropping for resolution standardization**
>
> We do perform image cropping for resolution standardization during data collection, as described in Section 2 (Data Collection). The detailed preprocessing statistics are provided in Appendix Table 4. Additionally, our preprocessing code is publicly available to provide complete implementation details.
>
> **(3) Caption counting methodology**
>
> To ensure fair comparison with existing grounding datasets (DIOR-RSVG, OPT-RSVG), Figure 3(d) statistics include only **phrase-level and sentence-level captions**. We exclude word-level captions from this distribution because:
>
> - Word-level captions (single category names) can have numerous random combinations. Including all possible word-level caption combinations would inflate the statistics and make them incomparable to existing RSVG benchmarks that focus on phrase and sentence descriptions.
> - During training, word-level captions follow the same category sampling approach as open-vocabulary aerial detection, where positive and negative categories are sampled for each training sample (detailed in Appendix E.1.3).
>
> ---

---

### Official Review · Reviewer_X2to · 2025-10-30

**Soundness:** 2
**Presentation:** 3
**Contribution:** 3
**Rating:** 6
**Confidence:** 3

**Summary:**

The paper proposes an automatic labeling engine, OS-W2S, which leverages large vision-language models (e.g., InternVL) with structured preprocessing and BERT-based postprocessing to generate word-, phrase-, and sentence-level annotations. Using this system, the authors construct MI-OAD, a large-scale language-guided open-set aerial detection dataset with 163K images and 2M image-caption pairs. The paper demonstrates substantial performance gains on three tasks (language-guided open-set detection, open-vocabulary detection, and remote-sensing visual grounding) achieving +31.1 AP50 and +34.7 Recall@10 improvements for Grounding DINO.

**Strengths:**

1. Originality: While the core methodology reuses existing components (e.g., InternVL, BERT, Grounding DINO), the paper exhibits a creative integration of large vision-language models for automated annotation in the aerial domain, which is a relatively unexplored application area. The introduction of a multi-granularity labeling strategy also shows originality in problem formulation. The OS-W2S pipeline effectively extends the scope of open-set detection from object categories to natural-language-level semantics.

2. Quality: The technical quality of the work is strong. The pipeline is methodologically sound, carefully engineered, and empirically validated. The experiments are thorough, covering multiple downstream tasks, and results consistently support the paper’s claims. The reproducibility is high, with sufficient implementation details and dataset statistics provided.

3. Clarity: The paper is very clearly written and well-structured. The figures and diagrams are of high quality, particularly those explaining the annotation pipeline and dataset structure.

4. Significance: The proposed MI-OAD dataset and OS-W2S labeling engine have strong practical and community significance. They address a major bottleneck in open-set remote sensing(data scarcity)and could serve as a foundation for future multimodal research in aerial imagery. Although the theoretical innovation is limited, the impact on benchmark construction and large-scale data automation is substantial and could influence related domains.

This paper’s strength lies in clarity, execution quality, and domain-level contribution. It delivers a solid, reproducible, and impactful system that will benefit the community, even though it does not introduce fundamentally new learning mechanisms.

**Weaknesses:**

1.The core innovation of OS-W2S lies in the integration of existing components (InternVL, BERT, Grounding DINO) rather than the introduction of new learning mechanisms or optimization objectives. Unlike recent ICLR papers that advance representation learning, this work primarily presents an engineering system for data generation.

2.The paper reports aggregate metrics (AP50, Recall@10) but does not investigate failure modes or error patterns in generated annotations. This omission weakens the empirical rigor, especially for a system that depends heavily on VLM predictions.

3.Although 10K samples were manually verified, this represents only 0.5% of the dataset. The authors should consider statistical sampling or cross-annotation consistency tests to better quantify the overall label reliability.

4.The related work section does not clearly differentiate OS-W2S from prior automatic labeling frameworks, such as LabelAnything: Multi-Class Few-Shot Semantic Segmentation with Visual Prompts (ECAI 2025) , which also leverage LLM/VLM pipelines.
The work’s weaknesses are not in execution but in research framing and analytical depth. With more rigorous evaluation of label reliability, stronger theoretical motivation for learning representations, and detailed analysis of model behavior, the paper could evolve from a engineering contribution into a research study.

**Questions:**

1.Currently, OS-W2S is described as a deterministic pipeline integrating pretrained models. Could the authors clarify whether any learnable components or adaptive mechanisms are included in the labeling process (e.g., fine-tuning InternVL or learning confidence thresholds)?

2.You mention that 10K samples were manually checked. Could you please describe how these samples were selected (random, stratified by scene type, or class-balanced)? A more rigorous sampling or reliability metric could help validate the robustness of your dataset.

3.Aerial imagery often involves dense and overlapping objects, where language-based models may confuse spatial relationships. How does OS-W2S handle such ambiguity? For instance, when multiple small vehicles are present, how is phrase-level annotation disambiguated? Could you provide examples of typical failure cases and mitigation strategies?

4.Since OS-W2S relies on large VLMs, it would be helpful to know how hallucinated or semantically incorrect captions are detected or filtered out. How do you filter out noisy subtitles?

5.How does OS-W2S differ from other automatic labeling systems that combine LLMs and VLMs, such as LabelAnything ?

6.Could the authors discuss how OS-W2S might influence learned feature representations? For example, does the multi-granularity annotation improve feature disentanglement or visual-textual alignment in downstream models?

7.Can OS-W2S be easily generalized to other domains (e.g., medical imaging, autonomous driving) where labeling cost is also high? If so, what adaptations would be needed, for example, domain-specific vocabulary or hierarchical label templates?

---

> ### Author Response · Authors · 2025-11-21
> **Response to Question 1**
>
> Thank you for your valuable time and constructive feedback. We hope to address your main concerns below and will revise the paper accordingly if it is accepted. Looking forward to your further discussion!
>
> ---
>
> ### Response to Question 1
>
> > Question comment 1. Currently, OS-W2S is described as a deterministic pipeline integrating pretrained models. Could the authors clarify whether any learnable components or adaptive mechanisms are included in the labeling process (e.g., fine-tuning InternVL or learning confidence thresholds)?
>
> ---
>
> Thank you for this question. OS-W2S does not include learnable components or adaptive mechanisms. It is designed as a model-agnostic pipeline that does not depend on any specific VLM. InternVL can be seamlessly replaced with other VLMs without modifying the core pipeline structure. This design provides several key advantages:
>
> **1. Adaptability to Rapid VLM Development**: The VLM field is evolving rapidly, with new models continuously improving in capability. Our model-agnostic, training-free design allows the pipeline to benefit from future VLM advancements without requiring pipeline redesign or retraining.
>
> **2. Flexibility for Different Use Cases**: OS-W2S is straightforward to reproduce and deploy. Users can independently replace the VLM component or adjust other processing modules according to their specific requirements, without managing training procedures or learned parameters. For instance, researchers with limited computational resources can use smaller models, while those prioritizing higher accuracy can employ larger, more capable VLMs.
>
> ---

---

> ### Author Response · Authors · 2025-11-21
> **Response to Question 2 & Weakness 3**
>
> > Question comment 2.You mention that 10K samples were manually checked. Could you please describe how these samples were selected (random, stratified by scene type, or class-balanced)? A more rigorous sampling or reliability metric could help validate the robustness of your dataset.
> >
> > Weaknesses comment 3. Although 10K samples were manually verified, this represents only 0.5% of the dataset. The authors should consider statistical sampling or cross-annotation consistency tests to better quantify the overall label reliability.
>
> ---
>
> Thank you for this question. As described in Section 4, we employed a rigorous quality control approach combining manual verification with statistical sampling and cross-model validation.
>
> **Manual Verification (10K samples):**
>
> The 10,000 manually checked samples were selected using a **class-balanced stratified sampling strategy** while ensuring scene diversity to provide comprehensive coverage across all categories.
>
> We grouped the MI-OAD validation image-caption pairs by category and had five experts manually select approximately 100 high-quality pairs per category. For categories with limited samples, we included all available instances and redistributed the remaining quota proportionally among categories with more than 100 samples, resulting in approximately 10,000 samples total.
>
> To ensure scene diversity, we ensured these captions were sampled from different images whenever possible, except for rare categories where multiple samples from the same image were necessary to achieve adequate representation.
>
> **Statistical Sampling and Cross-Model Validation:**
>
> Beyond the 10,000 manually reviewed samples for benchmark construction, we conducted additional statistical sampling to quantify overall annotation quality. Specifically, we sampled 300 images (approximately 1,765 instances) from MI-OAD and performed cross-annotation consistency tests using two independent powerful VLMs: InternVL3-78B (a stronger open-source model) and GPT-4o-mini (a leading closed-source model). These models verified the accuracy of three VLM-generated attributes (color, geometry, and relative position). InternVL3-78B achieved 98.98% (color), 99.21% (geometry), and 97.90% (relative position), while GPT-4o-mini achieved 96.88%, 94.39%, and 96.92%, respectively. These consistently high accuracy rates provide quantitative evidence of annotation quality across the entire dataset.
>
> ---

---

> ### Author Response · Authors · 2025-11-21
> **Response to Question 3**
>
> > Question comment 3. Aerial imagery often involves dense and overlapping objects, where language-based models may confuse spatial relationships. How does OS-W2S handle such ambiguity? For instance, when multiple small vehicles are present, how is phrase-level annotation disambiguated? Could you provide examples of typical failure cases and mitigation strategies?
>
> ---
>
> Thank you for this important question. The OS-W2S Label Engine is designed to handle aerial scenes with varying object densities through an **instance-level annotation approach**. The key advantage is that caption generation for each instance remains independent of the total number of objects in the image.
>
> ### **Instance-Level Annotation Process:**
>
> Our pipeline operates at the instance level through three stages:
>
> **1. Data Preprocessing (per instance)**
>
> For each individual instance in the image, we independently extract:
>
> - **Reliable attribute priors** (category, size, absolute location) derived from detection annotations using rule-based methods.
> - **Instance region crops** based on bounding boxes, allowing the VLM to focus on specific target objects.
> - **Foreground region crops** to provide local context for inferring relative position attributes while reducing confusion from broader background noise.
>
> This preprocessing is performed independently for each instance, regardless of scene complexity or object density.
>
> Note: In the cropped images, the target object is highlighted with a red bounding box (as shown in Figure 2), enabling the VLM to clearly identify which object is being annotated even in dense or overlapping scenarios.
>
> **2. Instance-Level Caption Generation (per instance)**
>
> Using the preprocessed inputs, the VLM generates attributes and captions for each instance independently (as shown in Figure 2). Each object receives:
>
> - Six attributes: category, size, color, geometry, relative position, and absolute position.
> - Three sentence-level captions with varying detail levels.
> - Three phrase-level captions based on attribute combinations.
>
> **3. Data Postprocessing (establishing caption-instance associations)**
>
> Since multiple instances may share similar attributes (e.g., "small red car"), we establish caption-instance associations by computing compositional attribute similarity between captions and instances. This enables a single caption to correctly match multiple instances when appropriate (detailed in Appendix C).
>
> **Handling Dense Scenes and Ambiguity:**
>
> For dense scenes with overlapping objects, our instance-level approach provides several advantages:
>
> - **Scalability**: Whether an image contains 5 or 100 objects, each instance is processed independently with consistent quality.
> - **Disambiguation through attributes**: The six-attribute framework provides sufficient specificity. For example, when multiple small vehicles are present, they are distinguished by combinations of color, geometry, and relative/absolute position attributes.
> - **Contextual balance**: The foreground region cropping limits the spatial context to nearby objects, reducing confusion from distant similar objects and background interference while preserving essential spatial relationships.
>
> ### **Quantitative Error Analysis**
>
> To rigorously evaluate annotation quality, we conducted independent validation in Section 4 by sampling 300 images (containing approximately 1,765 instances) and assessing VLM-generated attributes using two powerful models: InternVL3-78B and GPT-4o-mini. Results demonstrate consistently high accuracy across all attributes: InternVL3-78B achieved 98.98% (color), 99.21% (geometry), and 97.90% (relative position), while GPT-4o-mini achieved 96.88% (color), 94.39% (geometry), and 96.92% (relative position).
>
> Following your suggestion, we conducted in-depth analysis of the remaining errors to identify failure patterns and mitigation strategies. We found that annotation failures primarily occur in images with severe motion blur and target imaging issues, where objects are barely distinguishable even to human annotators. These challenging visual conditions lead to VLM annotation failures despite our domain adaptation strategies.
>
> **Addressing Image Quality Issues**: To mitigate this concern, we have employed Q-Align [1], a robust image quality assessment model, to systematically evaluate each image in MI-OAD. Every image was assigned a numerical quality score and categorical classification into five levels: excellent (0.1%), good (33.0%), fair (43.1%), poor (21.7%), and bad (2.1%). Researchers can leverage these quality annotations to selectively filter the dataset based on their specific requirements. Both the assessment procedure and complete quality annotations will be made publicly available with the dataset release.
>
> ---
>
> **References**
>
> [1] Wu H, Zhang Z, Zhang W, et al. Q-Align: Teaching LMMs for Visual Scoring via Discrete Text-Defined Levels[C]//International Conference on Machine Learning. PMLR, 2024: 54015-54029.
>
> ---

---

> ### Author Response · Authors · 2025-11-21
> **Response to Question 4 & Weakness 2**
>
> > Question comment 4.Since OS-W2S relies on large VLMs, it would be helpful to know how hallucinated or semantically incorrect captions are detected or filtered out. How do you filter out noisy subtitles?
> >
> > Weaknesses comment 2.The paper reports aggregate metrics (AP50, Recall@10) but does not investigate failure modes or error patterns in generated annotations. This omission weakens the empirical rigor, especially for a system that depends heavily on VLM predictions.
>
> ---
>
> Thank you for raising this important concern. We acknowledge the challenge of VLM-generated hallucinations and have implemented multiple mechanisms to minimize this phenomenon and ensure annotation quality.
>
> ### **High-Quality Source Datasets:**
>
> We construct MI-OAD based on eight widely-used aerial object detection datasets, including VisDrone, DOTA, and DIOR. These datasets are standardized benchmarks within the aerial detection community, characterized by rigorous human annotation processes and high-quality detection labels. This ensures our foundation is built on verified annotations.
>
> ### **Minimizing Domain Gap and Hallucination:**
>
> To address the domain gap between VLMs trained on natural images and aerial imagery, we implement two key strategies:
>
> **1. Ensuring Attribute Correctness through Reliable Priors**
>
> We explicitly provide the VLM with three deterministic prior attributes per instance:
>
> - **Category**: Directly obtained from detection annotations
> - **Size**: Computed as the area ratio using bounding box dimensions
> - **Absolute Location**: Determined by coordinates using rule-based methods
>
> These priors reduce ambiguity and help the VLM generate reliable descriptions. The remaining three attributes (color, geometry, relative position) are generated by the VLM based on visual content, as they are relatively robust to domain shifts.
>
> **2. Optimizing Visual Input through Two-Stage Cropping**
>
> - **Instance Region Cropping**: Combined with category and size priors, this ensures the VLM focuses on the specific instance, enabling accurate generation of color and geometry attributes while minimizing misclassifications.
> - **Foreground Region Extraction**: The target is highlighted with a red box in a slightly expanded crop, providing essential local context for inferring relative position attributes while reducing confusion from background noise.
>
> ### **Quality Validation:**
>
> **1. Structured Output Validation**
>
> All VLM outputs follow a predefined JSON format validated through regular expression parsing, ensuring 100% parsing success and structural consistency.
>
> **2. Independent VLM-Based Quality Assessment**
>
> We sampled 300 images (approximately 1,765 instances) and validated VLM-generated attributes using two independent powerful models: InternVL3-78B and GPT-4o-mini. Results demonstrate consistently high accuracy:
>
> - **InternVL3-78B**: color (98.98%), geometry (99.21%), relative position (97.90%)
> - **GPT-4o-mini**: color (96.88%), geometry (94.39%), relative position (96.92%)
>
> These high accuracy rates provide quantitative evidence of effective hallucination mitigation.
>
> **3. Downstream Task Performance**
>
> MI-OAD's effectiveness is validated through substantial improvements on multiple benchmarks:
>
> - **Remote Sensing Visual Grounding (Table 2)**: State-of-the-art results on DIOR-RSVG and OPT-RSVG, with +12.52 points in Pr@0.9 on OPT-RSVG
> - **Open-Vocabulary Aerial Detection (Table 3)**: Notable gains across DIOR (+7.1 AP50), DOTA-v2.0 (+5.2 mAP), and LAE-80C (+2.5 mAP)
>
> These consistent improvements demonstrate that MI-OAD provides high-quality annotations enabling effective model training rather than introducing harmful noise.
>
> **4. Manual Verification of Test Set**
>
> We conducted rigorous manual reviews during test set construction to establish MI-OAD as a reliable benchmark.
>
> ---

---

> ### Author Response · Authors · 2025-11-21
> **Response to Question 5 & Weakness 4（1/2）**
>
> > Question comment 5. How does OS-W2S differ from other automatic labeling systems that combine LLMs and VLMs, such as LabelAnything ?
> >
> > Weaknesses comment 4. The related work section does not clearly differentiate OS-W2S from prior automatic labeling frameworks, such as LabelAnything: Multi-Class Few-Shot Semantic Segmentation with Visual Prompts (ECAI 2025) , which also leverage LLM/VLM pipelines. The work’s weaknesses are not in execution but in research framing and analytical depth. With more rigorous evaluation of label reliability, stronger theoretical motivation for learning representations, and detailed analysis of model behavior, the paper could evolve from a engineering contribution into a research study.
>
> ----
>
> Thank you for this insightful question. We appreciate the opportunity to clarify how OS-W2S differs fundamentally from Label Anything and other automatic labeling systems.
>
> ### **Comparison with Label Anything**
>
> **1. Task and Objective Differences**
>
> Label Anything addresses **few-shot semantic segmentation** on natural images. Given a set with visual prompts (points, bounding boxes, or masks), it segments corresponding objects in query images by learning generalizable representations from limited examples.
>
> OS-W2S tackles a different problem: **automatic textual annotation generation** for language-guided open-set aerial object detection. It aim to automatically generates rich textual descriptions at multiple granularity levels (words, phrases, and sentences), transforming semantically sparse detection datasets into language-rich grounding data.
>
> **2. Domain and Challenge Differences**
>
> Label Anything operates on natural images using the $COCO-20^{i}$ benchmark.
>
> OS-W2S specifically targets aerial imagery, which presents distinct challenges including small object scales, dense object layouts, substantial background interference, and significant domain gaps from the natural images on which vision-language models are predominantly trained.
>
> **3. Technical Approach Differences**
>
> Label Anything employs prototype-based learning with transformer architectures. It extracts class prototypes by fusing visual prompts with image features through cross-attention mechanisms, then uses these prototypes to guide query image segmentation.
>
> OS-W2S takes a different approach centered on leveraging large vision-language models through a structured pipeline:
>
> - **Preprocessing Stage**: Generates reliable attribute priors (category, size, absolute position) from detection annotations and employs two-stage visual cropping (instance region and foreground region) to provide context-aware visual guidance
> - **VLM Interaction**: Uses carefully designed structured prompts combined with prior knowledge to guide the VLM in generating accurate attribute annotations (color, geometry, relative position)
> - **Postprocessing Stage**: Establishes caption-instance associations based on attribute similarity, enabling multi-instance annotations per caption
>
> It provides a training-free, model-agnostic, and flexible annotation framework that can seamlessly integrate future VLM advancements.
>
> ---

---

> > ### Author Response · Authors · 2025-11-21
> > **Response to Question 5 & Weakness 4（2/2）**
> >
> > ###
> >
> > ---
> >
> > ### **Compare with other aerial automatic labeling systems**
> >
> > Following your suggestion, we have compared OS-W2S with existing automatic annotation pipelines for aerial imagery.
> >
> > **LAE-Label Engine (LAE-1M)**: Focuses on vocabulary expansion for open-vocabulary aerial detection. While effective at increasing category diversity, it remains confined to category-level semantics and does not provide the rich textual descriptions needed for language-guided open-set detection.
> >
> > **RSVG Dataset Pipelines (DIOR-RSVG, OPT-RSVG):** Employ hand-crafted rules and fixed templates for caption generation. These pipelines are predominantly built on a single base dataset (DIOR), focus exclusively on the referring expression comprehension task, and enforce a one-caption-one-instance constraint by filtering out complex scenes. Consequently, they suffer from limited scene diversity, limited caption diversity, single-instance annotations, and relatively small scale.
> >
> > **OS-W2S Label Engine**: Addresses these limitations through three key innovations:
> >
> > - **Rich Multi-Granularity Descriptions**: Leverages large vision-language models to generate diverse textual annotations at word, phrase, and sentence levels based on six interpretable attributes (category, color, size, geometry, relative position, absolute position). This provides richer semantic information than both category-level labels and template-based captions, ensuring comprehensive caption diversity.
> >
> > - **Diverse Scenes with Multi-Instance Support**: The preprocessing and postprocessing stages enable handling of arbitrary aerial scenes without filtering, preserving comprehensive scene diversity while supporting flexible multi-instance caption-box associations beyond the REC task constraint, thereby overcoming both the limited scene coverage and single-instance limitations of prior work.
> >
> > - **Large-Scale Data Foundation**: By applying OS-W2S to eight widely used aerial detection datasets, we construct MI-OAD with 163,023 images and 2 million image-caption pairs, approximately 40 times larger than existing aerial grounding datasets, establishing a substantial data foundation for language-guided open-set aerial detection.
> >
> > **Action**: Following your valuable suggestion, we have add a dedicated subsection (Section B.4: Multi-modal Data Labeling for Aerial Detection and Grounding) in the revised paper. This new section is highlighted in blue in the revised manuscript.
> >
> > ---

---

> ### Author Response · Authors · 2025-11-21
> **Response to Question 6 & Weakness 1**
>
> > Question comment 6.Could the authors discuss how OS-W2S might influence learned feature representations? For example, does the multi-granularity annotation improve feature disentanglement or visual-textual alignment in downstream models?
> >
> > Weaknesses comment 1.The core innovation of OS-W2S lies in the integration of existing components (InternVL, BERT, Grounding DINO) rather than the introduction of new learning mechanisms or optimization objectives. Unlike recent ICLR papers that advance representation learning, this work primarily presents an engineering system for data generation.
>
> ---
>
> Thank you for raising this concern. We would like to clarify that this is a dataset and benchmark paper submitted to the ICLR Datasets and Benchmarks Track. The ICLR 2026 Call for Papers explicitly lists "datasets and benchmarks" as a core submission area, with recent programs accepting several dataset-centric contributions [1-5]. Our primary contribution is establishing a novel annotation methodology and constructing a foundational benchmark to advance representation learning for language-guided open-set aerial detection.
>
> ### **Dataset and Benchmark Contributions:**
>
> **1. Multi-Granularity Annotation Framework**: We propose the OS-W2S Label Engine, an automated pipeline that leverages VLMs to generate annotations at three semantic levels: words, phrases, and sentences. This design enables exploration of how language representations at different granularities guide visual understanding in aerial imagery. The pipeline is reproducible on accessible hardware (8× RTX 4090 GPUs) and model-agnostic, supporting seamless integration of future VLM advancements.
>
> **2. Domain Adaptation Methodology**: We design preprocessing and postprocessing operations to mitigate the domain gap between natural images and aerial imagery. Our preprocessing includes instance-region cropping, foreground extraction, and attribute priors derived from predefined rules, providing adequate context for accurate VLM annotations (color, geometry, relative position). Postprocessing then establishes caption-instance associations for multi-instance scenarios. These domain-specific adaptations address unique challenges in aerial imagery annotation.
>
> **3. Benchmark and Task Formulation**: We establish the first benchmark for language-guided open-set aerial detection with standardized evaluation protocols across multiple semantic granularities. This extends existing paradigms by requiring models to parse complex spatial relationships and learn fine-grained visual-semantic representations.
>
> **4. Empirical Analysis of Data Requirements**: MI-OAD (~2M grounding samples) provides a foundational resource for advancing this research area.
>
> ### **Alignment with ICLR's Focus on Representation Learning:**
>
> As the reviewer noted, by providing textual descriptions ranging from simple category names to complex sentences with spatial relationships, MI-OAD enables models to learn hierarchical visual-semantic representations at different granularities. For example, given an object "car," the model learns word-level alignment ("car"), phrase-level understanding ("a red car"), and sentence-level spatial reasoning ("a red car is parked on the right of the road, adjacent to a green taxi, located in the center of the image"). Tables 1-3 demonstrate that MI-OAD consistently improves model performance across open-vocabulary aerial detection, remote sensing grounding, and language-guided open-set aerial detection tasks.
>
> By advancing aerial detection to language-guided open-set detection, our work addresses real-world requirements and provides a benchmark for developing better visual-semantic representations.
>
> ---
>
> **References:**
>
> [1] Ran N, Xiao P, Wang Y, et al. HR-Extreme: A High-Resolution Dataset for Extreme Weather Forecasting[C]//ICLR. 2025.
>
> [2] Chen D, Huang Y, Wu S, et al. GUI-World: A Video Benchmark and Dataset for Multimodal GUI-oriented Understanding[C]//ICLR. 2025.
>
> [3] Shahreza H O, Marcel S. HyperFace: Generating Synthetic Face Recognition Datasets by Exploring Face Embedding Hypersphere[C]//The Thirteenth International Conference on Learning Representations.
>
> [4] Chen G, Liu Y, Huang Y, et al. CG-Bench: Clue-grounded Question Answering Benchmark for Long Video Understanding[C]//The Thirteenth International Conference on Learning Representations.
>
> [5] Zhang H, Wang X, Xu H, et al. MIntRec2. 0: A Large-scale Benchmark Dataset for Multimodal Intent Recognition and Out-of-scope Detection in Conversations[C]//The Twelfth International Conference on Learning Representations.
>
> ---

---

> ### Author Response · Authors · 2025-11-21
> **Response to Question 7**
>
> > Question comment 7.Can OS-W2S be easily generalized to other domains (e.g., medical imaging, autonomous driving) where labeling cost is also high? If so, what adaptations would be needed, for example, domain-specific vocabulary or hierarchical label templates?
>
> ---
>
> Thank you for this insightful question. While medical imaging and autonomous driving are not our primary research domains, we believe OS-W2S has potential for generalization to other specialized domains, such as underwater object observation.
>
> As the reviewer noted, generalizing OS-W2S to other domains would require domain-specific adaptations, including:
>
> - **Tailored attribute sets relevant to the target domain**: Different domains prioritize different attributes. In aerial imagery, we use six attributes (category, size, color, geometry, relative position, absolute position). For other domains, relevant attribute types should be extracted based on domain-specific characteristics.
> - **Adapted prompt engineering for VLM annotation**: The prompts guiding VLM annotation should be adapted to domain-specific knowledge.
>
> ---

---

> > ### Comment · Reviewer_X2to · 2025-11-25
> > **Comments on the authors' responses**
> >
> > Although the authors responded to my questions, these answers did not provide me with additional information about this work. I acknowledge that the experiments in this work are well-designed, demonstrating that the proposed method can be practically applied and establishing a corresponding dataset. However, I believe experiments constitute only part of a research effort; the authors should also explain in the paper why these designs were chosen and how they function. Currently, the paper only explains how these components are coupled and how automated data annotation is achieved step by step. These require not only experimental validation but also theoretical justification, clarifying the proposed method's architecture and its effectiveness. Without this theoretical framework, this work cannot be considered a complete scientific study but rather an engineering report intended to guide practical research. In my view, refocusing the paper's title or main content on the dataset itself might better align with its contributions than the current emphasis on the research problem of automated labeling.
> >
> > Therefore, I will maintain my rating but reduce my confidence to 2.

---

> > > ### Author Response · Authors · 2025-11-27
> > >
> > > Thank you for your continued engagement and for acknowledging the quality of our experimental design and practical applicability.
> > >
> > > 1. **Regarding Theoretical Framework for Pipeline Design**
> > >
> > > We agree that the theoretical justification for pipeline design is valuable and will become increasingly important as the field matures. However, we would like to note that VLM-based annotation pipelines in the current literature generally follow empirically-driven designs refined through careful prompt engineering and iterative experimentation. For instance:
> > >
> > > - **LAE-DINO [1]** develops a label engine that uses InternVL combined with rule-based filtering to expand categories for remote sensing detection.
> > > - **ShareGPT4V [2]** employs carefully designed data-specific prompts to interact with GPT-4V, generating 100K high-quality captions.
> > > - **MMDU [3]** uses clustering algorithms to select relevant Wikipedia content and designs prompt templates for GPT-4o to generate multi-turn dialogues.
> > >
> > > These representative works demonstrate that current VLM-based data annotation pipelines are predominantly built through iterative refinement with VLMs, where each component addresses specific practical challenges identified during development.
> > >
> > > In our paper, each component in OS-W2S has a clear and well-motivated purpose grounded in the specific challenges of aerial imagery. For example:
> > >
> > > - The **two-stage cropping strategy** addresses the difficulty of VLMs attending to small, densely packed objects in complex aerial backgrounds.
> > > - The **attribute prior injection** provides reliable prior knowledge for VLM interaction, enabling more accurate caption generation while minimizing hallucinations.
> > >
> > > These design rationales are validated through our quality assessment (>94% accuracy) and substantial downstream improvements. We are willing to add a dedicated appendix section that explicitly articulates the motivation behind each design choice if the paper is accepted.
> > >
> > > 2. **Regarding Dataset-Centric Contribution**
> > >
> > > We appreciate your suggestion to emphasize the dataset. We believe the OS-W2S Label Engine and MI-OAD dataset are complementary contributions that together lay the data foundation for language-guided open-set aerial detection. Our emphasis on the Label Engine stems from a key distinction: unlike previous automatic annotation works that rely heavily on powerful closed-source models (e.g., GPT-4), OS-W2S is designed to work with open-source VLMs on accessible hardware (8× RTX 4090 GPUs) while still achieving high-quality annotations. We hope this provides the remote sensing caption annotation community with a reproducible, low-cost, and low-barrier pipeline that can be widely adopted and extended.
> > >
> > > 3. **Regarding Scientific Research vs. Engineering Report**
> > >
> > > We acknowledge that dataset research, particularly pipeline-focused work, may appear to have less theoretical content compared to model-centric research. However, we respectfully suggest that dataset research serves as a critical foundation for subsequent theoretical advances. Constructing robust datasets and benchmarks is itself a scientific contribution. By establishing MI-OAD as a data foundation and benchmark for language-guided open-set aerial detection, we believe our work fulfills the core purpose of dataset research and enables future theoretical exploration in this direction.
> > >
> > > We sincerely thank you for the thoughtful discussion throughout this review process.
> > >
> > > References:
> > >
> > > [1] Pan J, Liu Y, Fu Y, et al. Locate anything on earth: Advancing open-vocabulary object detection for remote sensing community[C]//Proceedings of the AAAI Conference on Artificial Intelligence. 2025, 39(6): 6281-6289.
> > >
> > > [2] Chen L, Li J, Dong X, et al. Sharegpt4v: Improving large multi-modal models with better captions[C]//European Conference on Computer Vision. Cham: Springer Nature Switzerland, 2024: 370-387.
> > >
> > > [3] Liu Z, Chu T, Zang Y, et al. Mmdu: A multi-turn multi-image dialog understanding benchmark and instruction-tuning dataset for lvlms[J]. Advances in Neural Information Processing Systems, 2024, 37: 8698-8733.

---

### Official Review · Reviewer_eWjL · 2025-10-30

**Soundness:** 3
**Presentation:** 3
**Contribution:** 3
**Rating:** 4
**Confidence:** 2

**Summary:**

This paper presents an automatic labeling tool (OS-W2S) and uses it to create MI-OAD, a large-scale aerial imagery dataset designed to advance language-guided aerial open-set object detection research.

**Strengths:**

The paper is well organized.

The proposed MI-OAD dataset is a valuable, large-scale dataset for the community.

Experiments on YOLO-World and Grounding DINO demonstrated that MI-OAD can improve the model's performance in aerial object detection.

**Weaknesses:**

The core of the paper is to use VLM to generate annotations, but VLM itself may have biases (such as preferences for specific colors and shapes) and the risk of creating "illusions". Although the paper validates this with a stronger model in Section 4, "Quality Control Analysis", it does not fundamentally avoid the problem.

In section 5.4， Key terms like "OPT-RSVG" and "DIOR-RSVG" are not defined.

Why is the “Grounding DINO (+MI-PAD)” configuration in Table 2 much lower than the “LPVA” baseline under the cmuIoU metric?

**Questions:**

Please see the weakness.

---

> ### Author Response · Authors · 2025-11-21
> **Response to Weakness 1 (1/2)**
>
> Thank you for your valuable time and constructive feedback. We hope to address your main concerns below and will revise the paper accordingly if it is accepted. Looking forward to your further discussion!
>
> ---
>
>
> > Comment 1: The core of the paper is to use VLM to generate annotations, but VLM itself may have biases (such as preferences for specific colors and shapes) and the risk of creating "illusions". Although the paper validates this with a stronger model in Section 4, "Quality Control Analysis", it does not fundamentally avoid the problem.
>
> ---
>
> Thank you for raising this important point. Using VLMs for automated annotation significantly reduces manual effort and annotation costs, though this may introduce the risk of VLM hallucinations. Thus, we designed our annotation pipeline with careful **preprocessing strategies** to provide sufficient prior information and reduce model hallucinations before interfacing with the VLM. When the overall quality of the annotated dataset is ensured to be high, it can already bring substantial improvements to this field. Our independent validation using two stronger models (Section 4) demonstrates consistently high accuracy exceeding 94% across all attributes, confirming the high overall data quality. Furthermore, the substantial performance improvements on downstream tasks (Tables 2 and 3) demonstrate that the current annotation quality is sufficient for advancing existing benchmarks.
>
> We also conducted failure analysis and found that remaining errors primarily occur in low-quality images with severe motion blur. To address this, we **employed Q-Align [1] to assess and label the quality of each image**, allowing researchers to filter the dataset according to their specific requirements. Below, we detail our preprocessing strategies, validation results, and quality control measures.
>
>
> ### **Preprocessing Strategies to Mitigate VLM Hallucinations**
>
> **1. Ensuring Attribute Correctness through Reliable Priors**
>
> The quality of our annotations depends on the accuracy of six instance attributes: category, size, color, geometry, relative position, and absolute position. To reduce errors from the domain gap, we explicitly provide the VLM with three deterministic prior attributes per instance:
>
> - **Category**: Directly obtained from detection annotations
> - **Size**: Computed as the area ratio using bounding box dimensions
> - **Absolute Location**: Determined by coordinates using rule-based methods
>
> Providing these prior attributes reduces ambiguity and helps the VLM generate more reliable descriptions. The remaining three attributes (color, geometry, relative position) are generated by the VLM based on visual content, as they are relatively robust to domain shifts and can be accurately inferred from properly cropped regions.
>
> **2. Optimizing Visual Input through Two-Stage Cropping**
>
> Through iterative experimentation, we developed a two-stage cropping strategy that balances fine details and contextual information:
>
> - **Instance Region Cropping (Stage 1)**: We crop each object based on its bounding box and enlarge it by a scale factor (Figure 2). Combined with category and size priors, this ensures the VLM focuses on the specific instance, enabling accurate generation of color and geometry attributes while minimizing misclassifications.
> - **Foreground Region Extraction (Stage 2)**: We generate a slightly expanded foreground crop that includes the immediate surroundings of the target object, with the target highlighted by a red box. This provides essential local context for inferring relative position attributes while reducing confusion from background noise.

---

> > ### Author Response · Authors · 2025-11-21
> > **Response to Weakness 1 (2/2)**
> >
> > ### **Why This Approach Is Effective**
> >
> > Our strategy effectively addresses the domain gap and reduces VLM hallucinations by:
> >
> > - Using **instance-level crops** to guide the VLM's focus on fine-grained details
> > - Employing **foreground-level crops** to provide essential local context without excessive background interference
> > - Integrating reliable **prior attribute extraction** for attributes most sensitive to the domain gap (category, size, and absolute location), ensuring their accuracy through detection annotation priors
> >
> > For the remaining attributes (color, geometry, and relative position), which are relatively robust to domain shifts, the VLM provides accurate annotations. This combination further mitigates difficulties arising from the domain gap inherent to aerial imagery. The effectiveness of this approach aligns with recent research findings, including DAM [2], RegionGPT [3], and Alpha-CLIP [4], which demonstrate that region-aware input strategies enhance localized understanding.
> >
> > ### **Quality Validation**
> >
> > Beyond our preprocessing design, we ensure annotation quality through multiple validation approaches:
> >
> > **1. Structured Output Validation**
> >
> > All VLM outputs follow a predefined JSON format validated through regular expression parsing, ensuring 100% parsing success and structural consistency across all annotations.
> >
> > **2. Downstream Task Performance**
> >
> > MI-OAD's effectiveness is validated through substantial improvements on multiple benchmarks:
> >
> > - **Remote Sensing Visual Grounding (Table 2)**: State-of-the-art results on DIOR-RSVG and OPT-RSVG, with +12.52 points in Pr@0.9 on OPT-RSVG
> > - **Open-Vocabulary Aerial Detection (Table 3)**: Notable gains across DIOR (+7.1 AP50), DOTA-v2.0 (+5.2 mAP), and LAE-80C (+2.5 mAP)
> >
> > These consistent improvements demonstrate that MI-OAD provides high-quality annotations enabling effective model training rather than introducing harmful noise from VLM hallucinations.
> >
> > **3. Independent VLM-Based Quality Assessment**
> >
> > We sampled 300 images (approximately 1,765 instances) and validated VLM-generated attributes using two independent powerful models: InternVL3-78B (a stronger open-source model) and GPT-4o-mini (a leading closed-source model). Results demonstrate consistently high accuracy:
> >
> > - **InternVL3-78B**: color (98.98%), geometry (99.21%), relative position (97.90%)
> > - **GPT-4o-mini**: color (96.88%), geometry (94.39%), relative position (96.92%)
> >
> > These high accuracy rates provide quantitative evidence that our domain gap mitigation strategies effectively reduce VLM hallucinations.
> >
> > **Failure Analysis and Quality Control**
> >
> > We conducted in-depth analysis of the remaining errors to identify failure patterns. We found that annotation failures primarily occur in images with severe motion blur and target imaging issues, where objects are barely distinguishable even to human annotators. These challenging visual conditions lead to VLM annotation failures despite our domain adaptation strategies.
> >
> > **Addressing Image Quality Issues**: To mitigate this concern, we employed Q-Align [1], a robust image quality assessment model, to systematically evaluate each image in MI-OAD. Every image was assigned a numerical quality score and categorical classification into five levels: excellent (0.1%), good (33.0%), fair (43.1%), poor (21.7%), and bad (2.1%). Researchers can leverage these quality annotations to selectively filter the dataset based on their specific requirements. Both the assessment procedure and complete quality annotations will be made publicly available with the dataset release.
> >
> > ---
> >
> > **References:**
> >
> > [1] Wu H, Zhang Z, Zhang W, et al. Q-Align: Teaching LMMs for Visual Scoring via Discrete Text-Defined Levels[C]//International Conference on Machine Learning. PMLR, 2024: 54015-54029.
> >
> > [2] Lian L, Ding Y, Ge Y, et al. Describe anything: Detailed localized image and video captioning[J]. arXiv preprint arXiv:2504.16072, 2025.
> >
> > [3] Guo Q, De Mello S, Yin H, et al. Regiongpt: Towards region understanding vision language model[C]//Proceedings of the IEEE/CVF Conference on Computer Vision and Pattern Recognition. 2024: 13796-13806.
> >
> > [4] Sun Z, Fang Y, Wu T, et al. Alpha-clip: A clip model focusing on wherever you want[C]//Proceedings of the IEEE/CVF conference on computer vision and pattern recognition. 2024: 13019-13029.
> >
> > ---

---

> ### Author Response · Authors · 2025-11-21
> **Response to Weakness 2**
>
> > Comment 2: In section 5.4， Key terms like "OPT-RSVG" and "DIOR-RSVG" are not defined.
>
> ---
>
> Thank you for pointing this out. We apologize for the oversight. OPT-RSVG [1] and DIOR-RSVG [2] are existing remote sensing visual grounding benchmarks that we use to evaluate our dataset. Specifically:
>
> - **DIOR-RSVG**: A visual grounding dataset constructed from the DIOR aerial detection dataset, containing 17,402 images with 38,320 image-caption pairs.
> - **OPT-RSVG**: A larger visual grounding dataset combining three aerial detection datasets (DIOR, HRRSD, and SPCD), containing 25,452 images with 48,952 image-caption pairs.
>
> Both datasets are introduced in our Related Work (Appendix B.3) and compared with MI-OAD in Table 5 (Appendix). In the revised version, we will add brief definitions when these terms first appear in Section 5.4 to improve clarity. For example:
>
> *"We evaluate on two standard remote sensing visual grounding benchmarks: DIOR-RSVG [1] and OPT-RSVG [2]."*
>
> ---

---

> ### Author Response · Authors · 2025-11-21
> **Response to Weakness 3**
>
> > Comment3: Why is the “Grounding DINO (+MI-PAD)” configuration in Table 2 much lower than the “LPVA” baseline under the cmuIoU metric?
>
> ---
>
> Thank you for this question. We provide a detailed analysis below.
>
> **MeanIoU vs. cumIoU**
>
> Following existing RSVG evaluation protocols [1-2], we use two IoU-based metrics with different formulations:
>
> $$\text{meanIoU} = \frac{1}{N}\sum_{i=1}^{N}\frac{|G_i \cap P_i|}{|G_i \cup P_i|}$$
>
> $$\text{cumIoU} = \frac{\sum_{i=1}^{N}|G_i \cap P_i|}{\sum_{i=1}^{N}|G_i \cup P_i|}$$
>
> where $N$ is the number of samples, $G_i$ is the ground truth box, and $P_i$ is the top-1 predicted box. The key difference is that meanIoU treats all samples equally, while cumIoU weights contributions by box areas globally. Crucially, cumIoU is more sensitive to false detections: when a prediction misses the annotated target ($|G_i \cap P_i| = 0$), the denominator $|G_i \cup P_i|$ remains large, causing a heavier penalty than in meanIoU.
>
> The phenomenon we observe (strong meanIoU gains with modest cumIoU improvement) suggests our model mostly makes correct detections, though some cases do not align with DIOR-RSVG's specific annotations.
>
> **Root Cause**
>
> DIOR-RSVG follows the Referring Expression Comprehension (REC) paradigm, where each caption must correspond to exactly one object. MI-OAD adopts a multi-instance paradigm where captions can match multiple objects satisfying the description. This design better reflects real-world scenarios but creates evaluation mismatches with DIOR-RSVG.
>
> Through analysis of cases where $|G_i \cap P_i| = 0$, we identified three issues in DIOR-RSVG:
>
> **1. Ambiguous single-instance annotations.** Despite filtering images with more than 5 objects per category during construction, DIOR-RSVG still contains cases that match multiple objects. For instance, in image 01150.jpg, a caption is "the white airplane," yet four white airplanes appear in the scene, but only one is arbitrarily designated as ground truth for this caption. When our top-1 prediction is a different white airplane, cumIoU gets penalized even though the detection is correct.
>
> **2. Annotation errors.** Some DIOR-RSVG samples contain contradictory annotations. In image 01218.jpg, the caption states "the airplane is a little bigger than the gray airplane," but the image shows five gray airplanes arranged side by side, making the reference ambiguous and impossible to resolve correctly.
>
> **3. Complex linguistic references.** Some captions use confusing spatial descriptions that are difficult to parse. For example, in image 02826.jpg, the caption "the vehicle is on the right of the vehicle on the left" is ambiguous and difficult to interpret correctly.
>
> These issues cause many $|G_i \cap P_i| = 0$ cases, suppressing cumIoU growth. Moreover, even on DIOR-RSVG, Grounding DINO (+MI-OAD) substantially reduces the performance gap with LPVA compared to the baseline Grounding DINO (trained only on DIOR-RSVG train set), improving cumIoU from 79.36 to 81.69 (+2.33). This further validates MI-OAD's effectiveness.
>
> ---
>
> **References:**
>
> [1] Zhan Y, Xiong Z, Yuan Y. RSVG: Exploring data and models for visual grounding on remote sensing data. IEEE Transactions on Geoscience and Remote Sensing, 2023.
>
> [2] Li K, Wang D, Xu H, et al. Language-guided progressive attention for visual grounding in remote sensing images. IEEE Transactions on Geoscience and Remote Sensing, 2024.
>
> ---

---

### Official Review · Reviewer_RRL5 · 2025-10-31

**Soundness:** 2
**Presentation:** 2
**Contribution:** 3
**Rating:** 2
**Confidence:** 3

**Summary:**

The paper presents an automatic labeling pipeline for generating language-guided annotations in aerial imagery, and uses it to build a new large-scale dataset. The dataset integrates data from existing aerial detection sources and adds multiple types of text descriptions for each instance.  The authors show that models trained or adapted on MI-OAD achieve better performance on several aerial detection and grounding benchmarks.

**Strengths:**

- The paper tackles the lack of large-scale language-grounded datasets in the aerial domain, which is a real bottleneck for open-set detection research. MI-OAD is significantly larger and more diverse than existing aerial grounding datasets.
- Experiments are extensive and show clear improvements across several downstream benchmarks. The dataset and code are publicly released, making the work reproducible and potentially useful to the community.

**Weaknesses:**

- The discussion of related work focuses almost entirely on model architectures rather than dataset construction. Since this paper’s main contribution is a dataset and annotation pipeline, it should instead position the work within the context of existing dataset-building methodologies. A detailed quantitative comparison with prior aerial or language-grounded datasets is missing. The paper should explicitly articulate what is new about the proposed pipeline beyond scale, and how its annotation strategy differs from existing automatic labeling systems.

- The overall novelty is limited. The paper reads more like a detailed technical report that consolidates known components into a large dataset pipeline. While the engineering execution is solid, the scientific contribution is unclear. The paper lacks abstraction or theoretical framing that would justify it as a research advance suitable for ICLR. It would fit better as a data or resource paper for a vision-oriented venue such as CVPR / ICCV.

- An ablation analysis is missing. Since the annotation pipeline contains multiple preprocessing and postprocessing components, it would be important to remove or modify individual steps and evaluate model performance on the resulting datasets. This would verify that each design choice meaningfully contributes to dataset quality; otherwise, the complexity of the pipeline is not well justified.

- In the README.md source code (https://anonymous.4open.science/api/repo/MI-OAD/file/README.md)  of the provided code repository, a GitHub badge link (https://img.shields.io/github/stars/GT-Wei/MI-OAD) reveals the authors' identity. This appears to be unintentional, so I think maybe it is fine.

**Questions:**

See Weaknesses

---

> ### Author Response · Authors · 2025-11-21
> **Response to Weakness 1 (1/4)**
>
> Thank you for your valuable time and constructive feedback. We hope to address your main concerns below and will revise the paper accordingly if it is accepted. Looking forward to your further discussion!
>
> ---
>
> > Weaknesses comment 1. The discussion of related work focuses almost entirely on model architectures rather than dataset construction. Since this paper’s main contribution is a dataset and annotation pipeline, it should instead position the work within the context of existing dataset-building methodologies.
>
> ---
>
> Thank you for this valuable suggestion. We would like to clarify our organization below:
>
> (1) In the aerial detection and grounding domain, explicit work on automated dataset construction remains limited. We initially considered that existing work (LAE-DINO label engine and RSVG dataset construction pipelines) did not warrant a separate "dataset methodology" section, and instead discussed them within the corresponding related-work subsections (B.2 and B.3).
>
> (2) Our work also introduces a new task (language-guided open-set aerial detection) for the aerial domain. To help readers understand this task and its data requirements, we review existing aerial object detection and visual grounding work, explicitly identifying the scarcity of large-scale language-guided grounding data as the main bottleneck.
>
> (3) Following your suggestion, we have added a dedicated subsection (Section B.4: "Multi-modal Data Labeling for Aerial Detection and Grounding") in the revised manuscript that summarizes existing dataset construction methods and positions our OS-W2S Label Engine within this context. This new section is highlighted in blue.

---

> > ### Author Response · Authors · 2025-11-21
> > **Response to Weakness 1 (2/4)**
> >
> > > Weaknesses comment 1. A detailed quantitative comparison with prior aerial or language-grounded datasets is missing.
> >
> > ---
> >
> > Thank you for raising this point. We provide quantitative comparisons with prior datasets across the introduction, related work, and appendix. Below we summarize where these comparisons appear and what they demonstrate.
> >
> > **Introduction Comparisons**
> >
> > In the introduction, we contextualize our work by highlighting the scale of successful natural-image grounding datasets. We note that Grounding DINO v1.5 uses over 20 million grounding samples and DINO-X leverages over 100 million samples, illustrating the critical role of large-scale grounding data for effective open-set detection. We then systematically identify four key limitations of existing aerial grounding datasets: (i) limited scene diversity, (ii) limited caption diversity, (iii) single-instance annotations, and (iv) limited dataset scale. For each limitation, we explain how MI-OAD addresses it through our specific design choices.
> >
> > **Related Work Analysis**
> >
> > In Appendix B.3, we provide a detailed analysis of existing aerial grounding datasets and their construction pipelines. We discuss how current RSVG datasets (DIOR-RSVG, OPT-RSVG) rely on rule-based attribute extraction and fixed caption templates, and highlight the substantial scale gap between aerial and natural-image grounding data.
> >
> > **Appendix Comparisons**
> >
> > Table 4 summarizes the eight curated aerial detection datasets that serve as our foundation. Table 5 directly compares MI-OAD with existing remote sensing visual grounding datasets (RSVG-H, DIOR-RSVG, OPT-RSVG) across multiple dimensions including number of images, number of image-caption pairs, average caption length, and per-caption instance multiplicity. This comparison demonstrates that MI-OAD is approximately 40 times larger than comparable datasets while providing multi-instance, multi-granularity annotations.
> >
> > ---

---

> > > ### Author Response · Authors · 2025-11-21
> > > **Response to Weakness 1 (3/4)**
> > >
> > > > Weaknesses comment 1. The paper should explicitly articulate what is new about the proposed pipeline beyond scale.
> > >
> > > ---
> > >
> > > Thank you for pointing this out. We would like to clarify that the scale of MI-OAD represents only one aspect of our contribution. The key innovations of our work include:
> > >
> > > **1. Multi-Granularity Annotation Framework**: We propose the OS-W2S Label Engine, an automated pipeline that leverages VLMs to generate annotations at three semantic levels: words, phrases, and sentences. This design enables exploration of how language representations at different granularities guide visual understanding in aerial imagery. The pipeline is reproducible on accessible hardware (8× RTX 4090 GPUs) and model-agnostic, supporting seamless integration of future VLM advancements.
> > >
> > > **2. Domain Adaptation Methodology**: We design preprocessing and postprocessing operations to mitigate the domain gap between natural images and aerial imagery. Our preprocessing includes instance-region cropping, foreground extraction, and attribute priors derived from predefined rules, providing adequate context for accurate VLM annotations (color, geometry, relative position). Postprocessing then establishes caption-instance associations for multi-instance scenarios. These domain-specific adaptations address unique challenges in aerial imagery annotation.
> > >
> > > **3. Benchmark and Task Formulation**: We establish the first benchmark for language-guided open-set aerial detection with standardized evaluation protocols across multiple semantic granularities. This extends existing paradigms by requiring models to parse complex spatial relationships and learn fine-grained visual-semantic representations.
> > >
> > > **4. Empirical Analysis of Data Requirements**: MI-OAD (~2M grounding samples) provides a foundational resource for advancing this research area.

---

> > > > ### Author Response · Authors · 2025-11-21
> > > > **Response to Weakness 1 (4/4)**
> > > >
> > > > > Weaknesses comment 1. how its annotation strategy differs from existing automatic labeling systems.
> > > >
> > > > ---
> > > >
> > > > Thank you for this question. Existing aerial automatic labeling systems fall into two categories:
> > > >
> > > > 1. **LAE-Label Engine**: Focuses on vocabulary expansion for open-vocabulary aerial detection. While effective at increasing category diversity, it remains confined to vocabulary-level semantics and does not provide the rich textual descriptions needed for language-guided open-set detection.
> > > > 2. **RSVG Dataset Pipelines (DIOR-RSVG, OPT-RSVG)**: Use hand-crafted rules and fixed templates for caption generation. These pipelines are predominantly built on a single base dataset (DIOR), focus exclusively on the referring expression comprehension (REC) task, and enforce a one-caption-one-instance constraint by filtering out scenes with complex layouts. Consequently, they suffer from: (i) limited scene diversity, (ii) limited caption diversity, (iii) single-instance annotations, and (iv) relatively small scale.
> > > >
> > > > In contrast, our OS-W2S Label Engine addresses these limitations through several key design choices: (i) it leverages the natural language understanding capabilities of modern LVLMs to generate diverse, multi-granularity descriptions (word, phrase, and sentence levels) rather than template-based captions, (ii) its data preprocessing and postprocessing stages enable handling of diverse aerial scenes and support multi-instance caption-box associations beyond the REC task constraint, and (iii) its application to eight widely used aerial detection datasets achieves substantially larger dataset scale.
> > > >
> > > > ---

---

> ### Author Response · Authors · 2025-11-21
> **Response to Weakness 2**
>
> > Weaknesses comment 2. The overall novelty is limited. The paper reads more like a detailed technical report that consolidates known components into a large dataset pipeline. While the engineering execution is solid, the scientific contribution is unclear. The paper lacks abstraction or theoretical framing that would justify it as a research advance suitable for ICLR. It would fit better as a data or resource paper for a vision-oriented venue such as CVPR / ICCV.
>
> We respectfully disagree with this assessment. The ICLR 2026 Call for Papers explicitly lists "datasets and benchmarks" as one of its relevant subject areas, and we submitted our work under this area. In the current data-driven era, constructing high-quality datasets and establishing practical benchmarks represent important and valuable research contributions. This is precisely why major AI conferences including ICLR, NeurIPS, and ICML (not just vision-oriented venues like CVPR/ICCV) all explicitly include datasets and benchmarks in their scope.
>
> For dataset papers, the core contribution lies in constructing large-scale, high-quality datasets with minimal cost while demonstrating clear field advancement. We achieve this through: (1) carefully designed OS-W2S Label Engine enabling annotation on accessible hardware (8× RTX 4090 GPUs), (2) rigorous quality control including preprocessing and postprocessing strategies that enable VLMs to effectively handle the unique challenges of aerial imagery annotation, and (3) demonstrated substantial performance improvements on existing benchmarks while establishing the first benchmark for language-guided open-set aerial detection.
>
> These contributions align with ICLR's explicitly stated scope for dataset and benchmark papers.

---

> ### Author Response · Authors · 2025-11-21
> **Response to Weakness 3**
>
> > Weaknesses comment 3. An ablation analysis is missing. Since the annotation pipeline contains multiple preprocessing and postprocessing components, it would be important to remove or modify individual steps and evaluate model performance on the resulting datasets. This would verify that each design choice meaningfully contributes to dataset quality; otherwise, the complexity of the pipeline is not well justified.
>
> ---
>
> Thank you for this question. We would like to clarify that label engine pipelines differ from typical method papers where component ablations are standard practice. In OS-W2S, each component is specifically designed to address a distinct challenge in aerial imagery annotation, and these components work interdependently to enable the pipeline's operation. Removing any component would cause the pipeline to fail rather than simply degrade performance. Instead, we validate the pipeline's effectiveness through quantitative quality assessment and downstream task performance. Below, we provide detailed explanation of these considerations.
>
> **1. Component Interdependency and Design Rationale**
>
> Each component in OS-W2S serves a specific, essential purpose. For instance, the preprocessing module enables VLMs to effectively handle the unique challenges of aerial imagery annotation. Without preprocessing, VLMs would generate severe hallucinations due to the domain gap. Similarly, the postprocessing module is essential for establishing correct caption-instance correspondences. The role of each component is intuitive and directly observable during development, making their contributions evident without requiring ablation experiments.
>
> **2. Alternative Validation Approach**
>
> Instead of component ablations, we validate the pipeline's effectiveness through:
>
> - **Quantitative Quality Assessment (Section 4)**: We sampled and evaluated VLM-generated attributes using two independent models (InternVL3-78B and GPT-4o-mini), achieving consistently high accuracy across all VLM-annotated attributes. This directly demonstrates annotation quality.
> - **Downstream Task Performance**: As shown in Tables 1-3, MI-OAD enables substantial improvements across three tasks: language-guided open-set aerial detection, remote sensing visual grounding, and open-vocabulary aerial detection. These results demonstrate the effectiveness of both the OS-W2S Label Engine and the MI-OAD dataset.
>
> **3. Consistency with Prior Work**
>
> Prior label engine works in both aerial and natural image domains, such as LAE-DINO [1] for aerial detection and MMDU [2] for multi-turn dialog, also do not conduct component-wise ablations.
>
> **References:**
>
> [1] Pan J, Liu Y, Fu Y, et al. Locate anything on earth: Advancing open-vocabulary object detection for remote sensing community. AAAI 2025.
>
> [2] Liu Z, Chu T, Zang Y, et al. MMDU: A multi-turn multi-image dialog understanding benchmark and instruction-tuning dataset for LVLMs. NeurIPS 2024.
>
> ---

---

### Official Review · Reviewer_6Dr8 · 2025-11-02

**Soundness:** 3
**Presentation:** 3
**Contribution:** 3
**Rating:** 8
**Confidence:** 3

**Summary:**

The paper proposes  an automatic labeling engine that generates fine-grained textual annotations for aerial images using VLMs, and introduces MI-OAD, a large-scale dataset for language-guided openset aerial object detection. MI-OAD contains 163k images and 2 million image caption pairs at word, phrase, and sentence levels. Experiments show that pretraining on MI-OAD substantially boosts performance across open-vocabulary aerial detection, remote-sensing visual grounding, and zero-shot open-set detection tasks.

**Strengths:**

- Visual grounding has significant value and wide applications. Yet, existing dataset is not large enough to support the task. This paper proposed an automated way to generate grounding dataset using VLMs. The dataset will advance the research in this direction.

- The labeling pipeline is novel for aerial domains, combining structured preprocessing, VLM interaction, and BERT-based postprocessing. MI-OAD’s scale and multi-granularity annotation approach make it a comprehensive dataset for open-set aerial detection.

- Training or pretraining on MI-OAD significantly improves multiple benchmarks, indicating clear practical value and potential to establish a standard benchmark for language-guided aerial detection.

**Weaknesses:**

- Although sourced from eight aerial datasets, details about geographic, environmental, or temporal diversity are sparse. It is unclear whether MI-OAD adequately represents different regions, seasons, or sensor modalities.

- The label engine relies heavily on a single chosen VLM (InternVL-2.5-38B-AWQ), and the paper does not assess how dataset quality varies across models, e.g. evaluating usinng other VLMs.

- Only a very small portion of dataset is manually reviewed (0.5% of data).  The generalization of annotation quality to the remaining dataset is assumed, but not empirically proven.

- The paper emphasizes overall improvements but provides little qualitative or quantitative discussion of where MI-OAD annotations or trained models fail.

**Questions:**

- A more thorough analysis of the dataset, e.g. regions, sensor types, is encouraged.

- Alternative VLMs are encouraged to be analyzed for comparison.

- Additional human review is encouraged.

---

> ### Author Response · Authors · 2025-11-21
> **Response to Question 1 & Weakness 1**
>
> Thank you for your valuable time and constructive feedback. We hope to address your main concerns below and will revise the paper accordingly if it is accepted. Looking forward to your further discussion!
>
> > Question Comment 1: A more thorough analysis of the dataset, e.g. regions, sensor types, is encouraged.
> >
> > Weaknesses comment 1. Although sourced from eight aerial datasets, details about geographic, environmental, or temporal diversity are sparse. It is unclear whether MI-OAD adequately represents different regions, seasons, or sensor modalities.
>
> ---
>
> Thank you for raising this important point. We appreciate the opportunity to provide a more comprehensive analysis of the geographic, sensor, and temporal diversity represented in MI-OAD. Since some source datasets do not provide detailed statistics on all diversity dimensions, we report the information explicitly available in their original publications. Below, we detail the diversity characteristics across multiple dimensions.
>
> **Geographic Diversity**
>
> MI-OAD encompasses substantial geographic coverage spanning multiple continents and diverse landscape types. The constituent datasets collectively capture imagery from varied geographic contexts, including dense urban centers, suburban areas, rural regions, and natural landscapes. For instance, VisDrone was captured across 14 cities in China, covering metropolitan areas, suburbs, and traffic scenes with varying urban densities. xView provides particularly extensive coverage, spanning over 1,400 km² of satellite imagery from regions worldwide, including deserts, forests, plains, and coastal cities across different continents. SODA-A contributes unique coastal and Mediterranean landscapes through UAV imagery from 24 locations across the Maltese Islands. Other datasets such as DOTA, RSOD, and NWPU VHR-10 incorporate imagery from multiple international locations via satellite sensors and Google Earth, representing diverse settings including airports, harbors, industrial zones, residential areas, and coastal environments. This wide geographic coverage ensures MI-OAD represents diverse visual conditions and object appearances across different regions.
>
> **Sensor and Platform Diversity**
>
> The dataset incorporates imagery from multiple acquisition platforms:
>
> - Satellite platforms: WorldView-3 (xView), QuickBird, IKONOS (DOTA), and various Google Earth satellites
> - UAV platforms: DJI Mini 2, DJI Air 2S (SODA-A), and various drone models (VisDrone)
>
> **Temporal and Seasonal Diversity**
>
> MI-OAD captures diverse temporal and environmental variations across its constituent datasets. SODA-A explicitly covers seasonal variation from winter to spring (December 2022 to April 2023), while DIOR documentation specifically mentions diverse weather conditions. VisDrone includes images captured across varying times of day, weather conditions, and seasonal variations. Additionally, the combined datasets span from 2014 (NWPU VHR-10) to 2024 (SODA-A), reflecting the temporal diversity captured across multiple years of collection.
>
> ---

---

> ### Author Response · Authors · 2025-11-21
> **Response to Question 2 & Weakness 2**
>
> > Question Comment 2: Alternative VLMs are encouraged to be analyzed for comparison.
> >
> > Weaknesses comment 2. The label engine relies heavily on a single chosen VLM (InternVL-2.5-38B-AWQ), and the paper does not assess how dataset quality varies across models, e.g. evaluating usinng other VLMs.
>
> ---
>
> Thank you for this important question. The OS-W2S Label Engine is designed to be model-agnostic and can work with various VLMs. Our selection of InternVL-2.5-38B-AWQ was based on systematic evaluation across multiple leading models, balancing annotation quality with computational efficiency and accessibility. Below, we provide detailed rationale for our model selection and present comparative annotation results across multiple VLMs demonstrating consistent quality.
>
> **Rationale for Selecting InternVL-2.5-38B-AWQ**
>
> - **Model Evaluation**: During our preliminary assessment, we evaluated several leading VLMs, including Qwen2-VL and the InternVL2.5 series. Among all models tested, Qwen2-VL-72B, InternVL2.5-78B, and InternVL-2.5-38B (including their quantized versions) consistently achieved a 100% template-parsing success rate as verified by regular expression checks, demonstrating their ability to understand prompt guidance and reliably follow our structured output format. However, while larger models typically achieve higher annotation quality, they also demand greater computational resources.
> - **Efficiency and Cost Analysis**: InternVL-2.5-78B requires a minimum of **four 80GB GPUs** (4×A100), while InternVL-2.5-38B-AWQ can operate on **eight 24GB GPUs** (8×RTX 4090). In our trials, annotating 100 identical images took approximately 50 minutes for both configurations. Considering typical GPU rental prices (`4×A100 ≈ $4.18/h vs. 8×RTX4090 ≈ $2.09/h` ), InternVL-2.5-38B-AWQ reduces hardware costs by about half, thereby improving accessibility and reproducibility for the research community. Based on this balance of efficiency, cost, and scalability, we ultimately selected InternVL-2.5-38B-AWQ for our large-scale annotation.
>
> **Model-Agnostic Design**
>
> The OS-W2S Label Engine is designed as a **model-agnostic pipeline** that does not depend on any specific VLM. InternVL can be seamlessly replaced with other VLMs without modifying the core pipeline structure. This design provides flexibility for different use cases: researchers with limited resources can use smaller models, while those requiring higher accuracy can employ larger, more capable VLMs. The pipeline can also benefit from future VLM advancements, including emerging remote sensing-specialized models, without requiring pipeline redesign.
>
> **Cross-Model Quality Validation**
>
> Following your suggestion, We conducted additional validation using multiple VLMs to assess annotation consistency and quality:
>
> - **Independent Quality Assessment**: We sampled 300 images (approximately 1,765 instances) and validated VLM-generated attributes using two independent powerful models: InternVL3-78B and GPT-4o-mini. Results demonstrate consistently high accuracy:
>
>   - InternVL3-78B: color (98.98%), geometry (99.21%), relative position (97.90%)
>
>   - GPT-4o-mini: color (96.88%), geometry (94.39%), relative position (96.92%)
>
> - **Comparative Annotation Examples**: We also annotated sample images using different VLMs to demonstrate consistency. For instance, for image 06003.jpg from DIOR Val (a train station):
>
>   - **InternVL2.5-38B**: "a big train station with multiple tracks and platforms, situated adjacent to a curved road and surrounded by urban buildings, located at the center of the image."
>   - **InternVL2.5-78B**: "a big train station with multiple tracks and platforms, surrounded by urban buildings and a curved road, located at the center of the image."
>   - **InternVL3-78B**: "a big train station with multiple tracks and platforms, located adjacent to a busy urban area with buildings and roads nearby, situated at the center of the image."
>
>   Similarly, for image 07127.jpg (a stadium):
>
>   - **InternVL2.5-38B**: "a big stadium with a green field in the center, surrounded by parking lots and roads, located in the middle, left of the image."
>   - **InternVL2.5-78B**: "a big, oval-shaped stadium with a green field and blue seating, surrounded by trees and adjacent to a parking lot, located in the middle, left of the image."
>   - **InternVL3-78B**: "a big stadium with a green field and blue seating, surrounded by urban buildings and roads, located in the middle, left of the image."
>
>   These examples demonstrate that different VLMs produce semantically consistent and high-quality annotations, with variations primarily in descriptive details rather than fundamental accuracy. This consistency validates both our pipeline design and the quality of annotations generated by InternVL-2.5-38B-AWQ.
>
> ---

---

> ### Author Response · Authors · 2025-11-21
> **Response to Question 3 & Weakness 3**
>
> > Question Comment 3: Additional human review is encouraged.
> >
> > Weaknesses comment 3. Only a very small portion of dataset is manually reviewed (0.5% of data). The generalization of annotation quality to the remaining dataset is assumed, but not empirically proven.
>
> ---
>
> Thank you for this important comment. Language-guided open-set aerial detection is a data-driven task, and we have annotated over 2 million grounding samples in MI-OAD. However, incorporating manual verification throughout the construction process would require prohibitive costs at this scale. To address this challenge, we designed the OS-W2S Label Engine to leverage VLM capabilities for automated annotation while implementing a **multi-level quality assurance strategy** within this engine to ensure dataset quality.
>
> Our independent sampling validation demonstrates that VLM-generated annotations achieve over 94% accuracy across all evaluated attributes, confirming the quality of our annotation pipeline. Additionally, to establish MI-OAD as a reliable benchmark, we conducted comprehensive manual verification of the entire test set using stratified sampling for both class balance and scene diversity. Below, we detail each component of our quality assurance approach.
>
> #### Multi-Level Quality Assurance Strategy
>
> Our approach to ensuring annotation quality operates at several levels:
>
> **1. High-Quality Source Foundation**
>
> MI-OAD is constructed from eight widely-used aerial object detection datasets, including VisDrone, DOTA, and DIOR. These datasets are standardized benchmarks within the aerial detection community, characterized by rigorous human annotation processes and high-quality detection labels. This ensures our foundation is built on verified annotations.
>
> **2. Minimizing VLM Hallucinations**
>
> To address the domain gap between VLMs trained on natural images and aerial imagery, we implement two key strategies:
>
> - **Ensuring Attribute Correctness through Reliable Priors**:
>
>   We explicitly provide the VLM with three deterministic prior attributes per instance:
>
>   - **Category**: Directly obtained from detection annotations
>   - **Size**: Computed as the area ratio using bounding box dimensions
>   - **Absolute Location**: Determined by coordinates using rule-based methods
>
>   These priors reduce ambiguity and help the VLM generate reliable descriptions. The remaining three attributes (color, geometry, relative position) are generated by the VLM based on visual content, as they are relatively robust to domain shifts.
>
> - **Optimizing Visual Input through Two-Stage Cropping**:
>
>   - **Instance Region Cropping**: Combined with category and size priors, this ensures the VLM focuses on the specific instance, enabling accurate generation of color and geometry attributes while minimizing misclassifications.
>   - **Foreground Region Extraction**: The target is highlighted with a red box in a slightly expanded crop, providing essential local context for inferring relative position attributes while reducing confusion from background noise.
>
> **3. Systematic Quality Validation**
>
> - **Structured Output Validation**: All VLM outputs follow a predefined JSON format validated through regular expression parsing, ensuring 100% parsing success and structural consistency.
> - **Independent VLM-Based Quality Assessment**: We sampled 300 images (approximately 1,765 instances) and validated VLM-generated attributes using two independent powerful models: InternVL3-78B and GPT-4o-mini. Results demonstrate consistently high accuracy: InternVL3-78B achieved 98.98% (color), 99.21% (geometry), and 97.90% (relative position), while GPT-4o-mini achieved 96.88% (color), 94.39% (geometry), and 96.92% (relative position). These high accuracy rates provide quantitative evidence of effective hallucination mitigation.
> - **Downstream Task Performance**: MI-OAD's effectiveness is validated through substantial improvements on multiple benchmarks. For remote sensing visual grounding (Table 2), we achieve state-of-the-art results on DIOR-RSVG and OPT-RSVG, with +12.52 points in Pr@0.9 on OPT-RSVG. For open-vocabulary aerial detection (Table 3), we observe notable gains across DIOR (+7.1 AP50), DOTA-v2.0 (+5.2 mAP), and LAE-80C (+2.5 mAP). These consistent improvements demonstrate that MI-OAD provides high-quality annotations enabling effective model training rather than introducing harmful noise.
>
> **4. Rigorous Manual Verification of Test Set**
>
> We conducted comprehensive manual reviews during test set construction to establish MI-OAD as a reliable benchmark. The 10,000 manually verified samples were selected using a class-balanced stratified sampling strategy while ensuring scene diversity.
>
> ---

---

> ### Author Response · Authors · 2025-11-21
> **Response to Weakness 4**
>
> > Weaknesses comment 4. The paper emphasizes overall improvements but provides little qualitative or quantitative discussion of where MI-OAD annotations or trained models fail.
>
> ---
>
> Thank you for raising this important point. To address this concern, we provide detailed failure analysis below, including quantitative assessment of annotation errors and qualitative examination of challenging scenarios where trained models struggle.
>
> **Quantitative Error Analysis**
>
> We conducted independent quality assessment by sampling 300 images (approximately 1,765 instances) and validating VLM-generated attributes using two powerful models: InternVL3-78B and GPT-4o-mini. Results demonstrate consistently high accuracy: InternVL3-78B achieved 98.98% (color), 99.21% (geometry), and 97.90% (relative position), while GPT-4o-mini achieved 96.88% (color), 94.39% (geometry), and 96.92% (relative position).
>
> Following your suggestion, we conducted in-depth analysis of the remaining errors. We found that annotation failures primarily occur in images with severe motion blur and target imaging issues, where objects are almost indistinguishable to the naked eye. These challenging visual conditions lead to VLM annotation failures even with our domain adaptation strategies.
>
> - **Addressing Image Quality Issues:** To mitigate this concern, we have employed Q-Align [1], a robust image quality assessment model, to systematically evaluate each image in MI-OAD. Every image was assigned a numerical quality score and categorical classification into five levels: excellent (0.1%), good (33.0%), fair (43.1%), poor (21.7%), and bad (2.1%). Researchers can leverage these annotations to selectively filter the dataset based on their specific requirements. Both the assessment procedure and quality annotations will be made publicly available upon release.
>
> **Qualitative Analysis of Model Failures**
>
> We also examined failure cases in model outputs and identified challenging scenarios where trained models struggle. For example, models face difficulties detecting objects described through complex spatial relationships. Descriptions involving multiple relative position references, such as "a medium-sized pedestrian wearing a white top and dark pants is walking near a group of people and a bicycle, located in the lower middle, far left of the image," pose significant challenges for accurate localization. This suggests that models have difficulty simultaneously processing multiple spatial cues and compositional attribute descriptions.
>
> These analyses reveal an important research direction: better aligning spatial expressions in captions with implicit positional information in image features. Additionally, improving the alignment between multi-attribute descriptions and corresponding visual features could enhance language-guided open-set aerial detection performance. This represents a valuable avenue for future exploration.
>
> ---
>
> **References**
>
> [1] Wu H, Zhang Z, Zhang W, et al. Q-Align: Teaching LMMs for Visual Scoring via Discrete Text-Defined Levels[C]//International Conference on Machine Learning. PMLR, 2024: 54015-54029.
>
> ---

---

### Author Response · Authors · 2025-12-03

Dear PCs, SACs, ACs, and Reviewers,

Thank you for your valuable time and constructive feedback throughout the review process. To facilitate the review process for the newly assigned AC, we summarize our core contributions and the key discussion points below.

---

**Core Contributions:**

This paper introduces OS-W2S, a reproducible and accessible label engine, and establishes MI-OAD, the first foundational benchmark for language-guided open-set aerial detection. Our work addresses three key limitations in existing aerial detection paradigms:

1. **A New Task Formulation:** We introduce language-guided open-set aerial detection, which removes the constraint of predefined categories (compared to closed-set aerial detection), moves beyond vocabulary-level semantics (compared to open-vocabulary aerial detection), and overcomes the one-caption-one-object limitation (compared to remote sensing visual grounding), enabling comprehensive multi-granularity descriptions (from words to sentences) and one-caption-to-multiple-targets capability.

2. **OS-W2S Label Engine:** An automated pipeline generating multi-granularity annotations. It is reproducible, low-cost, training-free, and model-agnostic, requiring only 8×RTX 4090 GPUs and allowing seamless replacement of any other VLM.

3. **MI-OAD Benchmark:** The first benchmark in this domain, addressing the severe scarcity of grounding data with unprecedented scale (40× larger) and semantic richness, enabling practical language-guided open-set detection.

Reviewers consistently recognized these contributions. For example, Reviewer 6Dr8 noted MI-OAD will "advance the research in this direction," Reviewer X2to praised its "strong practical and community significance," and Reviewer RRL5 acknowledged it "tackles the lack of large-scale language-grounded datasets, which is a real bottleneck."

---

**Responses to Key Concerns:**

**1. Annotation Quality (Reviewers eWjL, X2to, zebP, 6Dr8):**

Our annotation quality is ensured through multiple complementary mechanisms: (1) high-quality source datasets providing a reliable foundation; (2) carefully designed preprocessing strategies that supply precise prior information to minimize domain gaps and reduce VLM hallucinations; (3) comprehensive quality validation through structured output checks, independent cross-model verification using InternVL3-78B and GPT-4o-mini (achieving over 94% accuracy), downstream task performance evaluation, and manual test set verification.

The effectiveness is directly demonstrated through substantial performance gains on established benchmarks.

- For open-vocabulary aerial detection, MI-OAD training yields +7.1 AP50 on DIOR, +5.2 mAP on DOTA-v2.0, and +2.5 mAP on LAE-80C.
- For remote sensing visual grounding, we achieve state-of-the-art results with consistent improvements: on OPT-RSVG, +6.89 Pr@0.5 and +6.95 meanIoU; on DIOR-RSVG, +4.61 Pr@0.5 and +4.55 meanIoU.

These consistent improvements confirm that MI-OAD advances the field with sufficient annotation quality.

**2. Complex Scene Annotation (Reviewers X2to, zebP):**

OS-W2S addresses the challenges posed by complex backgrounds and dense objects through an **instance-level annotation design**, where caption generation remains independent per instance, followed by post-processing that establishes caption-instance associations.

**3. Dataset Diversity (Reviewer 6Dr8):**

Geographically, MI-OAD spans from 14 Chinese cities (VisDrone) to Maltese coastal regions (SODA-A) and global satellite coverage (xView). Sensor platforms include satellites (WorldView-3, QuickBird, IKONOS) and UAVs (DJI series). Temporal coverage extends from 2014 to 2024 across different seasons and weather conditions.

**4. Scientific Value and Venue Appropriateness (Reviewers RRL5, X2to):**

ICLR 2026 explicitly lists "datasets and benchmarks" as a core submission area. Our paper introduces a practically-oriented task for the aerial domain, constructs a reproducible, low-cost, training-free, and model-agnostic label engine, and establishes the corresponding dataset and benchmark. Our work fulfills the core requirements for dataset contributions and provides a foundation for future theoretical exploration.

---

**Revisions Made:**

Based on the reviewers' suggestions, we have made the following enhancements to the manuscript:

(1) Adding a dedicated related work section (B.4) elaborating on distinctions from existing aerial annotation methods;

(2) Providing updated dataset statistics with both original and preprocessed image counts (Table 4);

(3) Incorporating image quality annotations via Q-Align, enabling researchers to flexibly filter the dataset according to their specific requirements.

---

We thank the reviewers and the AC for their feedback throughout this process. We believe this work will contribute meaningfully to advancing the aerial detection community.

Best regards,

The Authors

---

### Meta-Review · Area_Chair_VkMh · 2025-12-10

**Summary:**

This paper presents an automatic labeling engine that uses vision–language models (VLMs) to produce fine-grained textual annotations for aerial imagery and introduces MI-OAD, a large-scale dataset for language-guided open-set aerial object detection. Reviewers recognize the value of the newly collected dataset but raise critical concerns about annotation quality. Because annotations are generated by VLMs and evaluated using more powerful VLMs, the evaluation may be unreliable: both generator and evaluator models can share similar language biases and failure modes, especially in complex spatial relationships. Moreover, the proposed data-collection pipeline largely combines established processes and engineering heuristics, offering limited scientific novelty. Similar VLM-based data generation pipelines have already been explored in both machine learning and remote sensing communities, weakening the novelty claim.

**Reviewer Concerns:**

Across reviewers, major concerns include the reliability and scientific contribution of the proposed data-collection pipeline. The primary issue is annotation quality: labels generated by VLMs and evaluated by stronger VLMs may inherit similar language biases and failure modes, making the evaluation potentially unreliable, especially for complex spatial relationships or dense object distributions. Reviewers also question the limited novelty of the pipeline, noting that it largely combines established processes and resembles existing VLM-based data-generation frameworks in both the ML and RS communities. Additional concerns include the lack of detailed failure-case analysis, insufficient comparison with prior pipelines and datasets, and ambiguity in handling multiple objects of the same category. While authors provide extensive clarification for most of these questions, concerns regarding annotation reliability, evaluator bias, and scientific novelty remain outstanding.

**Reviewer Scores:**

Reviewer 6Dr8 gave an initial score of 8, with concerns about data diversity, evaluator VLM choice, annotation quality, and missing failure-case analysis. Most issues were addressed, though data quality concerns persist; the reviewer is likely to maintain a score of 8.
Reviewer RRL5 gave an initial score of 2, citing limited novelty, missing comparisons with earlier pipelines and datasets, and unclear contribution to a general ML conference rather than domain-specific venues. While the rebuttal clarified aspects of pipeline design, core concerns about novelty and contribution remain; the reviewer may keep the score at 2.
Reviewer eWjL gave an initial score of 4, focused on annotation quality and evaluator VLM bias. Clarifications did not fully resolve concerns about VLM–VLM evaluation reliability, and the reviewer may maintain the score at 4.
Reviewer X2to gave an initial score of 6, questioning the pipeline’s novelty, lack of failure-case analysis, label quality, and missing comparisons with existing VLM-based frameworks. Although quality control clarifications were helpful, evaluator reliability and ambiguity in dense-object scenarios remain unresolved. The reviewer indicated they will keep the score at 6, but with reduced confidence.
Reviewer zebP gave an initial score of 4, raising concerns about annotation quality and category unification. While clarifications were provided, doubts regarding VLM-based evaluation remain; the reviewer may keep the score at 4.

---

### Decision · Program_Chairs · 2026-01-26

Reject